# SELF-UPDATABLE LARGE LANGUAGE MODELS BY INTEGRATING CONTEXT INTO MODEL PARAMETERS

**Yu Wang**[1]*, **Xinshuang Liu**[1]*, **Xiusi Chen**[2], **Sean O'brien**[1], **Junda Wu**[1], **Julian McAuley**[1]

[1]University of California San Diego, [2]University of Illinois Urbana-Champaign

## ABSTRACT

Despite significant advancements in large language models (LLMs), the rapid and frequent integration of small-scale experiences, such as interactions with surrounding objects, remains a substantial challenge. Two critical factors in assimilating these experiences are (1) **Efficacy**: the ability to accurately remember recent events; (2) **Retention**: the capacity to recall long-past experiences. Current methods either embed experiences within model parameters using continual learning, model editing, or knowledge distillation techniques, which often struggle with rapid updates and complex interactions, or rely on external storage to achieve long-term retention, thereby increasing storage requirements. In this paper, we propose **SELF-PARAM** (Self-Updatable Large Language Models by Integrating Context into Model Parameters). SELF-PARAM requires no extra parameters while ensuring near-optimal efficacy and long-term retention. Our method employs a training objective that minimizes the Kullback-Leibler (KL) divergence between the predictions of an original model (with access to contextual information) and a target model (without such access). By generating diverse question-answer pairs related to the knowledge and minimizing the KL divergence across this dataset, we update the target model to internalize the knowledge seamlessly within its parameters. Evaluations on question-answering and conversational recommendation tasks demonstrate that SELF-PARAM significantly outperforms existing methods, even when accounting for non-zero storage requirements. This advancement paves the way for more efficient and scalable integration of experiences in large language models by embedding knowledge directly into model parameters. Code is open-sourced at https://github.com/XinshuangL/SELF-PARAM

## 1 INTRODUCTION

In dynamic environments, whether in virtual worlds such as video games or real-world human societies, developing a cognitive system capable of effectively interacting with objects poses significant challenges. We define the interactions between the cognitive system and its environment as experiences. To remain functional and adaptive, a cognitive system must continuously evolve by integrating new experiences and reflecting upon past interactions whenever engaged by the environment (Wang et al., 2024d). One of the primary challenges in this evolutionary process is the system's ability to absorb experiences, adapt accordingly, and recall key events in the future. This challenge highlights two critical properties that the system must possess: (1) **Efficacy**: The system must maintain an impeccable memory of recent events, ensuring that each step of knowledge integration is both accurate and effective. (2) **Retention**: The system should demonstrate a robust ability to recall experiences from the past, indicating strong long-term memory capabilities.

To achieve long-term Retention, a cognitive system must effectively store its experiences. Existing methods for managing past experiences can be categorized based on whether they require additional modules or parameters beyond the model's inherent parameters: (1) **Methods without Additional Modules or Parameters:** These approaches, including continual learning, model editing, and knowledge distillation techniques, embed knowledge directly into the model's parameters. For instance, model editing methods such as MEND (Mitchell et al., 2022), ROME (Meng et al., 2022), MEMIT (Meng et al., 2023), and Model-Editing-FT (Gangadhar & Stratos, 2024) update

---

*Y. Wang and X. Liu contribute equally; correspondence to: {yuw164,xil235}@ucsd.edu

specific components of the language model to incorporate or modify information. These methods are particularly effective for inserting factual statements like "The highest mountain above sea level is Mount Everest." However, they may be limited in handling more complex experiences that extend beyond simple facts. Continual learning methods (Sun et al., 2020a; Wu et al., 2024) can manage more intricate information but typically require compiling a corpus for pre-training or fine-tuning, which may not be suitable for frequent and rapid updates. Moreover, continual learning often relies on Next-Word Prediction (NWP) loss, which may not guarantee strong Efficacy in immediate knowledge integration. Existing Knowledge Distillation methods mainly focus on distilling factual statements (Padmanabhan et al., 2024) or prompts (Choi et al., 2022) into model parameters, which may fall short when injecting complicated experiences. (2) **Methods with Additional Modules or Parameters**: These approaches employ external components such as text repositories, constructed knowledge bases, or memory modules to store past experiences, thereby facilitating long-term Retention. For example, MemoryLLM (Wang et al., 2024c) and Retrieval-Augmented Generation (Lewis et al., 2020) allow models to access external information during inference. The language model can directly attend to external modules, which can enhance Efficacy by providing accurate and relevant information. However, relying on external storage necessitates maintaining large modules to store all past experiences, thereby increasing storage requirements. Smaller external modules usually offer only marginal benefits (Wang et al., 2024c; Bulatov et al., 2022; 2023).

To address these challenges, we propose a novel method that requires zero additional parameters while yielding nearly optimal Efficacy and maintaining robust long-term Retention. We call this method **SELF-PARAM** (Self-Evolvable Large language models with Parameter Integration). We formally define **context injection** as effective when the model can accurately answer questions about specific contexts after incorporating them into its parameters. Based on this definition, we introduce a training objective that minimizes the Kullback-Leibler (KL) divergence between the predictions of an original model (which has access to the context) and a target model (which does not). The intuition behind using KL divergence is that the conditional probabilities of the original model capture the relationships between future queries and the context. If our training sentences are sufficiently diverse (ideally encompassing a wide range of possible queries), the target model should be able to generate appropriate responses to any queries related to the context after training. Since it is infeasible to use all possible sentences for training, we employ an instruct model to generate question-answer pairs about the context, along with sentences sampled from the pre-training dataset, to construct a target sentence set for each context. By minimizing the KL divergence between the conditional distributions of the original model (conditioned on the context) and the predicted distributions of the target model (without the context), we effectively update the target model to internalize the context within its parameters.

To demonstrate the efficacy of our method, we evaluate SELF-PARAM on tasks involving Single Context Injection and Batch Context Injection. To assess knowledge retention, we conduct experiments on Sequential Context Injection. Additionally, we apply our method to a Conversational Recommendation task, injecting conversations between users and the system into an instruct model. In this setting, the models' recommendation recalls improve significantly over all other baselines. These comprehensive experiments show that SELF-PARAM surpasses existing methods by a large margin while maintaining zero additional parameters. This advancement effectively achieves both high Efficacy and robust long-term Retention without the need for extra storage modules.

## 2 RELATED WORK

Based on their extra storage needs, existing approaches for injecting context into models can be categorized into methods without extra storage and those with extra storage. We provide a detailed overview of each category below.

**Methods without extra storage**. This category primarily encompasses Model Editing, Continual Learning, and Knowledge Distillation techniques. For model editing methods (Yao et al., 2023), we focus on approaches that directly modify the model parameters, thereby embedding new knowledge within the parameters themselves. MEND (Mitchell et al., 2022) introduces rank-one model edits to update the model parameters, utilizing meta-learning to train a lightweight meta-network for the editing process. Similarly, methods such as ROME (Meng et al., 2022) and MEMIT (Meng et al., 2023) propose closed-form solutions for editing the MLP layers. In contrast, Model-Editing-FT (Gangadhar & Stratos, 2024) fine-tunes the entire model on the factual statements that need to

be injected. T-Patcher (Huang et al., 2023) and CaliNet (Dong et al., 2022) store new knowledge in additional neurons. However, these approaches may face challenges related to the continuous growth of parameters when the knowledge to be injected comprises lifespan experiences. As for continual learning, it is usually used in the following situations: (1) Real-Time Assimilation of Dynamic Data: Continual learning facilitates the integration of information from diverse sources such as news (Sun et al., 2020b), scholarly articles Cossu et al. (2022), and social media Cossu et al. (2022). (2) Injection of Extensive Knowledge into Language Models: Jang et al. (2022) presents a method for continual knowledge learning that effectively updates LLMs with new data without compromising previously acquired knowledge. (3) Domain Adaptation: Continual training on new data streams enables models to adapt to specific domains in both language and vision. Cossu et al. (2022) demonstrate this by continuously training models on evolving datasets. Additionally, Ke et al. (2023) propose a soft-masking mechanism to update language models with domain-specific corpora, enhancing performance while preserving general knowledge. Various domain-adapted models have been developed, including those tailored for the financial domain (Xie et al., 2023), E-commerce (Ma et al., 2023), and academic content (Wei et al., 2023). (4) Instruction Tuning for Enhanced Reasoning and Interaction: To enable models to tackle novel tasks, task-incremental continual instruction tuning has been proposed (Wang et al., 2024b; 2023). These tasks encompass mathematical problem-solving (Azerbayev et al., 2023), utilization of calculators, search engines, and databases (Qin et al., 2023). With the rapid development of new tools such as advanced software libraries, novel APIs, and domain-specific utilities (Liang et al., 2023; Jin et al., 2023), there is an increasing necessity to continuously update language models to swiftly adapt and master these innovations. In the realm of Knowledge Distillation, Choi et al. (2022) propose a method for distilling prompts directly into model parameters, effectively replicating scenarios where prompts are appended prior to generation. Later, Padmanabhan et al. (2024) introduces a technique for distilling factual knowledge into model parameters such as "ChatGPT is an AI chatbot developed by OpenAI." Both approaches demonstrate impressive performance in knowledge injection. However, they may not be directly applicable to the injection of experiences which can be much more complicated.

**Methods with extra storage**. While the preceding category primarily focus on updating the model itself with new knowledge, another line of research involves storing knowledge in external modules and leveraging these modules to generate responses as needed. This approach encompasses several methodologies: (1) Retrieval-Augmented Generation: This framework involves maintaining a repository of past knowledge, which can be either in the form of knowledge graphs or organized texts (Wang et al., 2024d), from which relevant information is extracted using a retrieval mechanism when generating responses (Gao et al., 2023; Karpukhin et al., 2020; Sarthi et al., 2024; Gutiérrez et al., 2024; Izacard et al., 2022; Wang et al., 2022). (2) Long Context Methods: These methods store all relevant knowledge within the model's context window. When prompted, the model processes both the extensive context and the input prompt to generate an appropriate response (Wang et al., 2024a; Tworkowski et al., 2023; Chen et al., 2023; Han et al., 2023). (3) Memory-Augmented Methods: This approach typically involves maintaining a memory pool where knowledge is stored. These methods generally include two main operations: (i) Write Operation: Updating the memory pool, often through the compression of textual information into the memory space (Zhou et al., 2023; Wang et al., 2024c). (ii) Read Operation: Retrieving relevant information from the memory pool to inform response generation (Wang et al., 2024c; Zhong et al., 2023). Although memory-augmented methods can achieve impressive performance, they may require significant additional storage space to accommodate complex knowledge externally. In contrast, SELF-PARAM circumvents the need for external storage by embedding all information directly into the model weights via backpropagation. This approach eliminates the additional space requirements associated with external modules, allowing the model to retain and utilize knowledge internally without external dependencies.

## 3 METHODOLOGY

### 3.1 DEFINITION OF CONTEXT INJECTION

This paper introduces the concept of **Context Injection** in language models. We define *Context Injection* as the process of modifying a language model to incorporate specific context $x$, enabling the model to generate accurate and context-aware responses to related queries. Let $\theta$ represent the original language model that does not contain the information in $x$. When querying the original model $\theta$ with a prompt $p$ related to the context $x$, the generated answer $a_1$ is given by:

$$a_1 = \texttt{Generate}(\theta, p).$$

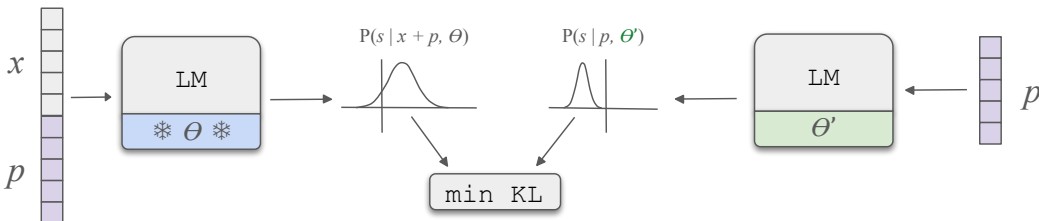

Figure 1: **The Process of Context Injection.** Given the original model $\theta$ and a context $x$, our goal is to obtain a new model $\theta'$ that incorporates $x$ directly into its parameters. To achieve this, we require that for any prompt $p$, the model $\theta'$ generates the same output as the original model $\theta$ would when provided with the combined input $x + p$. In other words, $\theta'$ should emulate $\theta$'s behavior with access to the context $x$ when only prompted by $p$. For a random sentence $s$, let $P(s \mid x + p, \theta)$ denote the token distribution generated by the original model $\theta$ when prompted with $x + p$, and let $P(s \mid p, \theta')$ represent the distribution from the new model $\theta'$ when prompted with $p$. Ideally, for all possible sentences $s$, we aim to minimize the KL divergence between $P(s \mid x + p, \theta)$ and $P(s \mid p, \theta')$. This ensures that $\theta'$ accurately integrates the context $x$ within its parameters.

For example, consider the following context and prompt:

$$x : \texttt{A national survey} \cdots \texttt{ only 38\% willingly} \cdots$$

$$p : \texttt{What percentage of bank branches willingly disclose fees} \cdots \texttt{?}$$

Here we omit some words with "$\cdots$" to save space. In this scenario, the answer $a_1$ produced by $\theta$ is likely to be incorrect due to the absence of relevant information in the model. However, by incorporating the context $x$ into the prompt, the model can generate a more accurate response:

$$a_2 = \texttt{Generate}(\theta, x + p),$$

where $x + p$ denotes the concatenation of $x$ and $p$ into a single, extended prompt. Assuming that $\theta$ is sufficiently robust and the necessary information is contained within $x$, the resulting answer $a_2$ should be correct. Then we define a new model $\theta'$ as the obtained model after injecting the context $x$ into the original model $\theta$. When $\theta'$ is queried with the prompt $p$, the generated answer $a_3$ is:

$$a_3 = \texttt{Generate}(\theta', p).$$

If $a_3$ is accurate, this indicates that we have successfully incorporated the context $x$ into the original model $\theta$. In summary, *Context Injection* involves updating the original language model $\theta$ to $\theta'$ by embedding a specific context $x$, thereby enabling the model to accurately respond to any questions related to $x$. This holds true provided that the original model $\theta$ can generate correct answers when supplied with both the context $x$ and the relevant question.

This definition leads to the first essential property of Context Injection: **Efficacy**. Efficacy is achieved when the modified model $\theta'$ can accurately answer questions pertaining to the injected context $x$. Furthermore, consider a sequence of contexts $x_1, \ldots, x_n$ injected sequentially into $\theta$ to obtain a final model $\theta_n$. **Retention** is defined as the degree to which $\theta_n$ preserves the information from previously injected contexts $x_1, \ldots, x_{n-1}$. Retention can be measured by querying $\theta_n$ with questions related to each $x_i$ for $i \in \{1, \ldots, n-1\}$. In this paper, our objective is to maximize both efficacy and retention while maintaining a storage complexity $\mathcal{S}$ of zero.

### 3.1.1 THE PROCESS OF CONTEXT INJECTION

To integrate a context $x$ into the model parameters, we aim to ensure that the updated model $\theta'$ generates responses identical to those produced by the original model $\theta$ when the latter is provided with the combined prompt $x + p$. Specifically, the original model $\theta$ utilizes the context $x$ when responding to the prompt $p$. This approach is illustrated in Figure 1. Formally, to enable $\theta'$ to respond accurately to queries about $x$, we seek to approximate the following relationship:

$$\texttt{Generate}(\theta', p) \approx \texttt{Generate}(\theta, x + p), \tag{1}$$

where $p$ represents a prompt related to the context $x$, such as a question pertaining to $x$. Achieving an ideal scenario where Equation Eq. 1 holds for any possible prompt $p$ implies that any question

answerable by $\theta$ with the context $x + p$ can also be addressed by $\theta'$ using only $p$. This signifies effective context injection. Consequently, we define the following optimization objective:

$$\theta' = \arg\min_{\hat{\theta}} \mathbb{E}_s \Big[ KL \big[ P_\theta(s \mid x, p) \,\|\, P_{\hat{\theta}}(s \mid p) \big] \Big], \tag{2}$$

where $s$ denotes any sentence, whether related or unrelated to $x$, and $KL[\cdot\|\cdot]$ represents the KL divergence. If the objective in Equation Eq. 2 is minimized to zero, then Equation Eq. 1 is inherently satisfied. This concept is depicted in Figure 1. In practice, it is exhaustive to collect all prompts, thus we simply absorb the prompt into $s$ to obtain the following objective:

$$\theta' = \arg\min_{\hat{\theta}} \mathbb{E}_s \Big[ KL \big[ P_\theta(s \mid x) \,\|\, P_{\hat{\theta}}(s) \big] \Big], \tag{3}$$

Moreover, aggregating the KL divergence across all possible sentences $s$ is infeasible. Therefore, we employ sampling techniques to approximate the objective effectively. The sampling methodology for $s$ is detailed in the subsequent section. We denote the set of sentences used to train $\theta'$ according to Eq. 3 as $\mathcal{E}$ and we call this set as **Target Sentence Set** in the remaining paper.

## 3.2 CONSTRUCTION OF **TARGET SENTENCE SET**

To absorb $x$ into $\theta$, we need to construct the target sentence set which contains two parts: (1) sentences related to $x$ (to inject the context $x$) and (2) sentences unrelated to $x$ (to maintain the model's original abilities). As for (1), we employ an instruct model (we tried two settings: using `gpt-4o-mini` and using the corresponding instruct model, i.e., `Mistral-7B-Instruct-v0.3` for `Mistral-7B` and `Llama-3-8B-Instruct` for `Llama-3-8B`) to generate a set of contextually relevant question-answer pairs as the sentence set. The detailed prompt used for constructing is shown in Appendix A.3. Then for (2), we randomly sample sentences from SlimPajama dataset (Soboleva et al., 2023) as the unrelated set of sentences. Note that SlimPajama helps preserve the model's general capabilities, effectively acting as a form of regularization during training. With the balanced sentences from the context-related and context-unrelated datasets, we hope to inject the specialized knowledge encapsulated in $x$ into the model while maintaining the model's broader linguistic capabilities.

## 3.3 DISCUSSION

**Where does the performance gain come from?** We argue that the new objective in Eq. 3 encourages more effective knowledge injection than simply using the next-word prediction (NWP) loss that is widely used in continual learning (Zhang et al., 2023). In our experiments (shown in Table 1 and Table 2), we implement the baseline of fine-tuning the model on the context and then querying it with the question Q (denoted as "FT (C), Q"), where the results show that the model fine-tuned with NWP loss yields poor performance when responding to queries, indicating a failure in effective context injection (i.e., failure of Efficacy). To understand the inherent reason, consider the following example: let $x$ be a simple sentence "`David likes apples.`", a model trained with NWP loss on $x$ may overfit to the specific form of the question, failing to answer correctly when queried in a different format – for example, "`What fruit does David like?`". However, our method trains on various forms of question-answering pairs pertaining to the same underlying knowledge, which encourages the model to both absorb the knowledge and use it, instead of simply memorizing the facts in specific forms.

**Analysis of the Computational Cost** We acknowledge that our algorithm could introduce additional computation costs. Compared to the fine-tuning baseline, our approach roughly doubles the overall computational cost in terms of both time and monetary resources. Specifically, our method involves four key steps: (a) generating QA pairs, (b) computing base logits, (c) computing updated logits, and (d) performing backpropagation and optimization. In contrast, continual fine-tuning baselines, such as FT(C) and FT(S) in Tables 1-3, only require steps (c) and (d). Assuming each step is of comparable time complexity, our total cost is approximately twice that of continual fine-tuning. However, given the effectiveness of our method, we argue that these additional costs are justified.

**Hallucination Mitigation** As we are encouraging the model to generate answers without seeing the context, there is the risk of hallucination, where the model produces an answer even when it lacks the necessary knowledge. To mitigate this issue, we carefully filter out prompts that require

direct access to prior context in the original PwC dataset. For example, prompts such as *"summarize the previous context"* and *"write a title for the above text"* were removed, leaving only factual questions. This refined dataset contains context and questions that are based on constant knowledge, such as `Question: What language was spoken in the tape recording of Jamal Khashoggi's murder? Answer: Arabic`. These factual statements are less likely to induce hallucination, which helps support the method's robustness. Regarding the Target Sentence Set, we are not directly fine-tuning the model to predict each word within this set. Instead, we employ KL divergence to transfer the behavior from the original model to the updated one, which may also reduce the risk of hallucination.

## 4 EXPERIMENTS

In our experiments, we mainly consider the following tasks: (1) **Single Context Injection** (§ 4.1). We adopt SELF-PARAM to inject a single context into the model and then ask the model with the related questions. (2) **Batch Context Injection** (§ 4.2). We adopt SELF-PARAM to inject multiple contexts simultaneously and then ask the model with the questions related to the injected contexts. (3) **Sequential Injection** (§ 4.3). We sequentially inject multiple contexts into the model, ensuring that the model does not retain access to earlier contexts during the injection of newer ones. We then test the model's ability to recall and respond to questions related to the previously injected contexts, thereby assessing its long-term retention. (4) **Conversational Recommendation**(§ 4.4). We inject up to 1,000 conversations between users and the system into the model to allow the model to read the conversations. After injection, we prompt the model to generate movie recommendations and calculate the recall of these recommendations to measure the recommendation quality. In these tasks, (1), (2), (4) correspond to **Efficacy** evaluation, while (3) represents the **Retention** evaluation.

### 4.1 SINGLE CONTEXT INJECTION

**Experimental Setup** We use PwC dataset (Ge et al., 2023), consisting of triples in the form `(context, question, answer)`. To ensure concise answers, we filter out examples where the answers exceed five tokens, resulting in 1,581 unique contexts and 3,353 unique questions. From these, we select the first 100 contexts paired with 225 questions for the single context injection task. For each `(context, question, answer)` example, we inject the `context` into the model and then query the obtained model with the corresponding `question`. We evaluate performance using the QA-F1 score, as defined in LongBench (Bai et al., 2023). We conduct experiments with three backbone models: OpenLLaMA-3B-v2 (Geng & Liu, 2023), Mistral-7B (Jiang et al., 2023), and Llama3-8B (Dubey et al., 2024). We compare our proposed method, SELF-PARAM, against the following baselines: (1) **Base**: The original, unmodified model. (2) **FT (C)**: Fine-tuning the original model solely on the `context`. (3) **FT (S)**: Fine-tuning the original model on the target sentence set.

**Overall Performance Comparison** The results are summarized in Table 1. Our method, SELF-PARAM, significantly outperforms the fine-tuning baselines, demonstrating its effectiveness in knowledge injection. Specifically, fine-tuning the model solely on the context (**FT (C), Q**) yields poor performance, albeit slightly better than the unmodified base model (**Base, Q**). This indicates that while some knowledge is injected, much of it remains inaccessible. The poor performance of **FT (C), Q** highlights a major limitation

|  | Openllama-3B-v2 | Mistral-7B | Llama3-8B |
|---|---|---|---|
| Base, C+Q | 0.5043 | 0.5461 | 0.5368 |
| Base, Q | 0.1122 | 0.1503 | 0.1051 |
| FT (C), Q | 0.2217 | 0.2433 | 0.1213 |
| FT (S), Q | 0.2742 | 0.3246 | 0.3115 |
| SELF-PARAM, Q | **0.4927** | **0.3705** | **0.4213** |

Table 1: Experimental results for Single Context Injection. "C+Q" indicates that both `context` and `question` are provided as input to the model, serving as an upper bound for injection performance. "Q" denotes that only the `question` is provided, requiring the model to answer the questions based solely on its internal parameters.

of continual learning approaches that rely on NWP loss: they suffer from low **Efficacy**, as evidenced by the model's inability to generalize knowledge across different query formulations. In contrast, SELF-PARAM trains the model using diverse question-answer pairs related to the same context, which encourages the model to both absorb and utilize the knowledge effectively. This approach prevents overfitting to specific question formats, ensuring that the model can handle varied queries

about the injected context. Moreover, **FT (S), Q** performs worse than SELF-PARAM, underscoring that our method does more than merely injecting generated QA pairs. Instead, SELF-PARAM facilitates the model's ability to internalize and apply the context dynamically. This observation aligns with our training objective in Eq. 3, where minimizing the KL divergence between the original and target models allows SELF-PARAM to better approach the upper bound represented by **Base, C+Q**.

## 4.2 BATCH CONTEXT INJECTION

**Experimental Setup**  We use the PwC dataset and the same three backbone models as in § 4.1. The preprocessing is the same. Then we extract 100 and 500 contexts, with 225 questions and 1044 questions, respectively, from the obtained subset to perform batch injection. The injection step and the evaluation process are separated. We first inject 100/500 `context` into the model, and then we ask the model all `questions`. Note that one `context` may have more than one `question`, we calculate the average f1-score of multiple questions for one context and then calculate the average f1-score across all contexts. We conduct experiments on the same three backbone models as in § 4.1: OpenLLaMA-3B-v2 (Geng & Liu, 2023), Mistral-7B (Jiang et al., 2023), and Llama3-8B (Dubey et al., 2024). We compare our proposed method, SELF-PARAM, against the following baselines: (1) **Base**, the original, unmodified model; (2) **FT (C)**, fine-tuning the original model solely on the `contexts`; (3) **FT (S)**, fine-tuning the original model on question-answering pairs generated by an instruct model; (4) **MemoryLLM-8B**, a memory-augmented method continually trained on Llama3-8B with an additional memory module of size 1.67B tokens (Wang et al., 2024c); (5) **InfLLM**, a long-context method that can be integrated with large language models to achieve an effectively infinite context window; (6) **Dense Passage Retrieval (DPR)**(Karpukhin et al., 2020), a retrieval-augmented generation (RAG) method that encodes documents and questions into embeddings and uses `faiss` for document retrieval; (7) **BM25**, a sparse retriever-based RAG method; and (8) **RAPTOR**(Sarthi et al., 2024), which constructs knowledge graphs using an advanced language model (we use `gpt-4o-mini`) and leverages the knowledge graph for retrieval. For the long-context methods, we concatenate all `contexts` to form a single, extended context input and prompt the model with: "$context_1, \cdots, context_N, question$". For RAG methods (DPR and BM25), we treat each `context` as an individual document and perform document-level retrieval. Specifically, we use top-4 retrieval for OpenLLaMA-3B-v2, top-8 retrieval for Mistral-7B and Llama3-8B, and top-1 retrieval for RAPTOR.

Here we introduce the concept of Storage Complexity $\mathcal{S}$ as proposed in Wang et al. (2024d), which indicates the relative storage space required to store all the contexts that need to be injected compared to the size of all the contexts (such as the total number of tokens in all contexts). As we do not require any additional storage space, SELF-PARAM has $\mathcal{S}$ as zero. In contrast, RAG and long-context methods need to store all contexts, indicating $O(n)$ storage complexity. As MemoryLLM has a fixed-sized memory, the storage complexity is $O(1)$. Here for each baseline, we mark the storage complexity to indicate the general complexity which could also be applied to similar methods, rather than the specific complexity for every single baseline, thus offering a more generalized perspective.

**Overall Performance Comparison**  The results are summarized in Table 2. Our method, SELF-PARAM, consistently outperforms all baselines across different models and context sizes, demonstrating its superior ability to inject and utilize multiple contexts without requiring additional parameters. From Table 2, it is evident that SELF-PARAM consistently achieves the highest QA-F1 scores across all models and context sizes. Specifically, SELF-PARAM closely approaches the upper bound performance of **Base, C+Q** without requiring any additional parameters. This demonstrates the method's ability to effectively inject and utilize multiple contexts solely through parameter updates. In contrast, the fine-tuning baselines (**FT (C)**, **FT (S)**) exhibit significantly lower performance, especially as the number of contexts increases. Notably, **FT (C)**, which relies on next-word prediction (NWP) loss, struggles with maintaining **Efficacy** in knowledge injection, similar to observations in single context injection. Meanwhile, **FT (S)** performs better than **FT (C)** but still falls short of SELF-PARAM, highlighting that our method does more than merely injecting generated QA pairs—it facilitates the model's ability to internalize and apply the context dynamically. This observation aligns with our training objective in Eq. 3, where minimizing the Kullback-Leibler (KL) divergence between the original and target models allows SELF-PARAM to approach the upper bound represented by **Base, C+Q**. Memory-augmented methods like **MemoryLLM-8B** and retrieval-augmented generation (RAG) methods such as **DPR**, **BM25**, and **RAPTOR** require additional parameters or modules to store contexts, leading to increased storage complexity ($O(n)$).

| # of Contexts | Openllama-3B-v2 | | Mistral-7B | | Llama3-8B | | $\mathcal{S}$ |
|---|---|---|---|---|---|---|---|
| | 100 | 500 | 100 | 500 | 100 | 500 | |
| Base, C+Q | 0.5043 | 0.4869 | 0.5461 | 0.5725 | 0.5368 | 0.5089 | - |
| Base, Q | 0.1122 | 0.0814 | 0.1503 | 0.1382 | 0.1051 | 0.0967 | - |
| FT (C), Q | 0.2085 | 0.1231 | 0.1659 | 0.1433 | 0.1065 | 0.0878 | 0 |
| FT (S), Q | 0.1925 | 0.1784 | 0.2350 | 0.3459 | 0.3179 | 0.2848 | 0 |
| MemoryLLM-8B | - | - | - | - | 0.1435 | 0.0841 | $O(1)$ |
| InfLLM | 0.1437 | 0.1003 | 0.1619 | 0.1783 | 0.1301 | 0.1244 | $O(n)$ |
| DPR | 0.2795 | 0.2528 | 0.3175 | 0.3092 | 0.2184 | 0.2310 | $O(n)$ |
| BM25 | 0.1475 | 0.1872 | 0.3104 | 0.3135 | 0.3083 | 0.2862 | $O(n)$ |
| RAPTOR | 0.1344 | 0.1529 | 0.2133 | 0.1969 | 0.2000 | 0.2055 | $O(n)$ |
| SELF-PARAM, Q | **0.5082** | **0.5048** | **0.4521** | **0.4384** | **0.4368** | **0.4221** | 0 |

Table 2: Overall performance comparison on the task of batch injection. Here "C+Q" means providing the model with the specific context containing the answer for each question. Thus "Base, C+Q" serves as the upper bound.

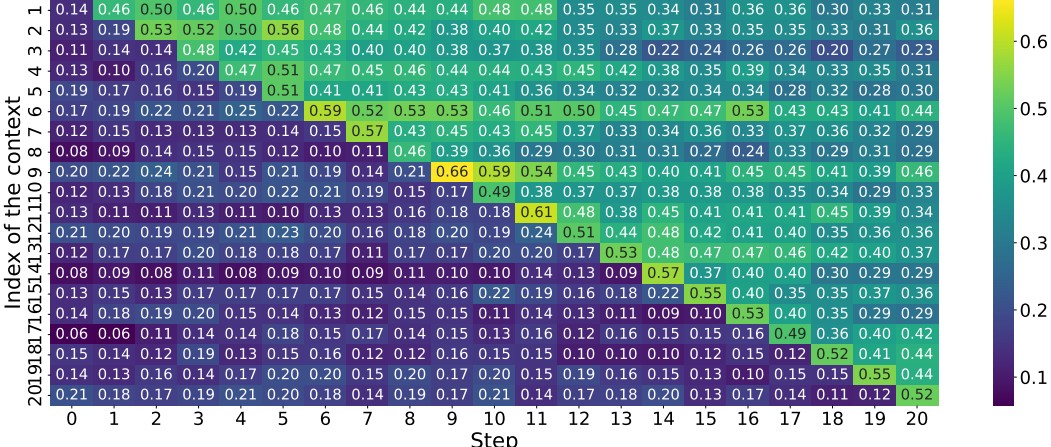

Figure 2: Average QA-F1 scores after sequentially injecting contexts into the model across 50 sequences. For each sequence, 20 contexts are injected one by one. The first column (step 0) represents the performance of the base model when queried without any injected context. Each subsequent column (step $i$, where $1 \leq i \leq 20$) shows the model's QA-F1 score on each of the contexts across all 20 contexts after $i$ injection steps. The displayed scores are the mean values averaged over all 50 sequences, demonstrating the model's retention ability as contexts are progressively injected.

Although some of these methods achieve competitive performance, they do not match the effectiveness of SELF-PARAM while incurring additional storage overhead.

## 4.3 SEQUENTIAL INJECTION

**Experimental Setup** To evaluate the model's ability to retain knowledge after multiple injections – termed Retention – we conduct sequential injection experiments. Specifically, we aim to determine whether the model can remember previously injected contexts after successive updates. We construct a list of 20 unique `contexts` from the PwC dataset (Ge et al., 2023) and inject them into the model one after another in a sequential manner. After each injection step, we assess the model's performance by calculating the QA-F1 score on all questions related to the injected contexts. The QA-F1 scores are averaged across all `questions` to obtain an average F1-score for each `context`. To ensure the robustness of our evaluation, we create 50 distinct sequences of `contexts`. We adopt the Mistral-7B model (Jiang et al., 2023) as our backbone model for these experiments. For each injection step within a sequence, we fine-tune the model using Low-Rank

| Model | INSPIRED | | | | REDIAL | | | |
|---|---|---|---|---|---|---|---|---|
| | $r_1$ | $r_2$ | $r_3$ | $r_4$ | $r_1$ | $r_2$ | $r_3$ | $r_4$ |
| Base | 0.0277 | 0.0277 | 0.0356 | 0.0316 | 0.0316 | 0.0293 | 0.0333 | 0.0312 |
| FT (C) | 0.0000 | 0.0000 | 0.0000 | 0.0000 | 0.0000 | 0.0000 | 0.0000 | 0.0000 |
| FT (S) | 0.0000 | 0.0000 | 0.0000 | 0.0000 | 0.0000 | 0.0000 | 0.0000 | 0.0000 |
| DPR | 0.0277 | 0.0198 | 0.0277 | 0.0198 | 0.0337 | 0.0295 | 0.0350 | 0.0314 |
| BM25 | 0.0198 | 0.0158 | 0.0198 | 0.0158 | 0.0318 | 0.0289 | 0.0331 | 0.0306 |
| RAPTOR | 0.0198 | 0.0198 | 0.0356 | 0.0316 | 0.0324 | 0.0299 | 0.0343 | 0.0316 |
| SELF-PARAM | **0.0357** | **0.0316** | **0.0395** | **0.0357** | **0.0337** | **0.0310** | **0.0360** | **0.0326** |

Table 3: Recall@1 under four different scenarios. As described in § A.1.1, $r_1$, $r_2$, $r_3$, and $r_4$ correspond to No Filtering, Seen Items Filtered Only, OOV Items Filtered Only, and Both OOV and Seen Items Filtered, respectively. Detailed metric definitions are provided in Appendix A.1.1.

Adaptation (LoRA) weights. After each step, the LoRA weights are merged into the model to incorporate the new context. The results are depicted in Figure 2, illustrating the QA-F1 scores after each sequential injection step. We also include the ablation studies in Appendix B.2

**Results and Discussions** As shown in Figure 2, our method, SELF-PARAM, demonstrates retention capabilities. Initially, after the first injection, the QA-F1 scores are high on the questions related to the context injected just now. As the number of injections increases to 20 steps, the QA-F1 scores gradually decline to approximately 0.3. Despite this reduction, SELF-PARAM maintains significantly higher performance compared to the base model, which exhibits a QA-F1 score of approximately 0.14 without any injected contexts. The diagonal of the figure shows that our model retains its functionality even after multiple rounds of injection, indicating our model's robustness.

### 4.4 CONVERSATIONAL RECOMMENDATION

**Experimental Setup** Building upon the previous sections where knowledge injection was performed using textual `contexts`, we explore injecting conversational data into the model. Specifically, we adopt the **Conversational Recommendation** task, which involves multi-turn interactions between users and a recommendation system. The intuition is that exposure to such conversations enables the model to generate more accurate and user-aligned movie recommendations. We utilize two datasets: (1) **INSPIRED** (Hayati et al., 2020): Contains 731 conversational interactions. (2) **REDIAL** (Li et al., 2018): Comprises 7,415 conversational interactions. Following the evaluation procedure from He et al. (2023), we prompt the target instruct model to generate 20 movie recommendations based on user queries. The recommendations are then evaluated using recall metrics ($r_1, r_2, r_3, r_4$), detailed in Appendix A.1.1. Our task formulation involves injecting a substantial number of conversations (731 for INSPIRED and 1,000 for REDIAL) into the model's parameters using SELF-PARAM, aiming to enhance recommendation quality compared to the backbone model without access to these conversations. We use the **Mistral-7B-instruct-v0.2** model as our backbone. We compare SELF-PARAM against the following baselines: (1) **FT (C)**: Fine-tuning the model on conversations, with loss calculated solely on system utterances; (2) **FT (S)**: Fine-tuning the model on generated question-answer pairs, with loss calculated only on answers; (3) **Dense Passage Retrieval (DPR)** (Karpukhin et al., 2020): Encodes conversations and queries into embeddings, retrieving relevant conversations using `faiss`; (4) **BM25**: A sparse retriever-based method for retrieving relevant conversations; (5) **RAPTOR** (Sarthi et al., 2024): Constructs knowledge graphs from conversations using `gpt-4o-mini` and retrieves relevant information from the graph. For retrieval-augmented generation (RAG) methods (DPR, BM25, RAPTOR), all conversations are stored in a knowledge pool. During inference, the most relevant conversations are retrieved and incorporated into the prompt as context.

**Overall Performance Comparison** The results are presented in Table 3. SELF-PARAM consistently outperforms all baselines across both datasets, demonstrating its effectiveness in enhancing recommendation quality without requiring additional storage modules. From Table 3, the following observations can be made: (1) SELF-PARAM achieves the highest Recall@1 scores across all scenarios and datasets, outperforming all baselines. This underscores the effectiveness of injecting conversations directly into the model parameters, enabling better understanding and recommenda-

tion generation. (2) Both **FT (C)** and **FT (S)** exhibit zero performance across all scenarios, indicating that traditional fine-tuning methods fail to effectively integrate conversational knowledge. This may be due to the divergent styles between recommendation conversations and the original instruct models, leading to degradation in model behavior. However, unlike fine-tuning baselines, SELF-PARAM maintains consistent performance without disrupting the model's inherent capabilities. This is attributed to the use of KL divergence in our training objective, which aligns the target model's behavior with the backbone model while integrating new knowledge. (3) Methods like DPR, BM25, and RAPTOR, which rely on external retrieval mechanisms, achieve little or no improvements. This may be due to the difficulty in effectively retrieving and integrating relevant conversations from a large pool, limiting their ability to enhance recommendation quality. Overall, SELF-PARAM demonstrates significant improvements in recommendation accuracy by effectively internalizing conversational knowledge without additional storage overhead. This highlights the potential of SELF-PARAM in enhancing interactive and dynamic tasks within large language models.

## 4.5 ABLATION STUDY

**Ablation Study of Training Objective** Since we introduce a new training objective, as described in Section 3.1.1, it is also important to examine the performance when applying Next-Word-Prediction loss directly to either the original context or the constructed target sentence set. To explore this, we include two baseline settings, FT (C) and FT (S), in Tables 1, 2, and 3, which correspond to fine-tuning on the original contexts and the target sentence set, respectively. The results show that while fine-tuning on the target sentence set (FT (S)) generally outperforms fine-tuning on the original contexts (FT (C)), it still falls significantly short of our proposed approach. This highlights the importance of our training objective in achieving superior performance.

**Ablation Study of the Model for Target Sentence Set Construction** To demonstrate that our performance gains do not stem from using a stronger model – and to distinguish our approach from knowledge distillation methods that transfer knowledge from larger models to smaller ones – we conduct an experiment where we replace the Target Sentence Set generated by `gpt-4o-mini` (as described in Section 4.2) with a cor-

|  | 100 Contexts | | 500 Contexts | |
|---|---|---|---|---|
|  | Mistral-7B | Llama3-8B | Mistral-7B | Llama3-8B |
| `4o-mini` | **0.4521** | **0.4368** | 0.4384 | 0.4221 |
| `instruct` | 0.4502 | 0.4341 | **0.4836** | **0.4464** |

Table 4: Ablation Study of the Model For Target Sentence Set Construction. Here `4o-mini` refers to `gpt-4o-mini`, `instruct` means using the corresponding instruct model, i.e., using Mistral-7B-Instruct-v0.3 for Mistral-7B and Llama-3-8B-Instruct for Llama3-8B.

responding instruct model. This ensures that no additional knowledge from a more advanced model is introduced, aligning with our "self-updatable" objective. The results, presented in Table 4, confirm that our method remains effective even without leveraging a more powerful model to construct the target sentence set. This further validates the robustness of our approach.

## 5 CONCLUSION AND FUTURE WORK

In this paper, we addressed the critical challenge of integrating small-scale experiences into large language models (LLMs) with high efficacy and long-term retention without incurring additional storage complexity. We introduced SELF-PARAM (Self-Updatable Large Language Models with Parameter Integration), a novel method that embeds experiences directly into model parameters by minimizing the Kullback-Leibler (KL) divergence between an original model with context and a target model without context. Through comprehensive evaluations across diverse tasks, including Single Context Injection, Batch Context Injection, Sequential Injection, and Conversational Recommendation, we find that SELF-PARAM consistently outperformed existing baselines, demonstrating superior performance in both immediate knowledge integration and robust retention over multiple injections. Notably, SELF-PARAM achieved these advancements without requiring additional parameters or external storage modules, thereby maintaining the model's integrity and scalability in dynamic environments. These findings highlight the potential of SELF-PARAM to enhance the adaptability and efficiency of LLMs. Future work may explore scaling SELF-PARAM to larger models, incorporating more diverse types of experiences, and applying the method to a broader range of applications such as sessions of conversations.

## ETHICS STATEMENT

In this research, we use publicly available datasets (PwC, INSPIRED, and REDIAL) that do not contain personally identifiable information, ensuring compliance with data privacy standards and licensing agreements. By embedding contextual knowledge directly into model parameters, there is a potential risk of data leakage; to address this, we adhere to robust security practices and recommend further safeguards for deployment. All methodologies and data handling procedures comply with relevant legal frameworks and ethical guidelines, and we have maintained research integrity through transparent reporting and appropriate citations. Overall, while our approach enhances the capabilities of large language models, we remain committed to promoting fairness, security, and responsible innovation in artificial intelligence.

## REPRODUCIBILITY STATEMENT

We describe construction process of the target sentence set in § A.3.

**Single Context Injection** (§ 4.1): We describe the dataset setup in § 4.1 and describe the implementation details in § A.2.

**Batch Context Injection** (§ 4.2): Similarly, we describe the dataset setup in § 4.2 and describe the implementation details in § A.2.

**Sequential Injection** (§ 4.3): The dataset setup is described in § 4.3 and the implementation details are shown in § A.2.

**Conversational Recommendation** (§ 4.4): The dataset setup is described in § 4.4 and the implementation details are described in § A.2.

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

# A    ADDITIONAL SETTINGS

## A.1    ADDITIONAL EXPERIMENTAL SETUPS

### A.1.1    METRICS FOR CONVERSATIONAL RECOMMENDATION TASKS

Following He et al. (2023), we report Recall@1 using the code base from `https://github.com/AaronHeee/LLMs-as-Zero-Shot-Conversational-RecSys` under four distinct scenarios. These scenarios are defined based on the post-processing steps applied to the generated recommendations, focusing on the filtering of Out-Of-Vocabulary (OOV) items and Seen items. Below, we describe each metric concisely:

**Metric Descriptions**

- $r_1$: No Filtering Applied
  - **OOV Items Filtered?** ✗
  - **Seen Items Filtered?** ✗
  - **Description**: In this baseline scenario, no post-processing filters are applied. All recommended items are included regardless of whether they are out-of-vocabulary (OOV) or have been previously mentioned in the current conversation. This metric serves as an upper bound for recommendation performance, reflecting the model's raw ability to generate relevant suggestions without any constraints.
- $r_2$: Seen Items Filtered Only
  - **OOV Items Filtered?** ✗
  - **Seen Items Filtered?** ✓
  - **Description**: This scenario filters out items that have already been mentioned in the ongoing conversation (Seen Items) while retaining all OOV items. By eliminating redundant recommendations, $r_2$ ensures that the user receives new and potentially more relevant suggestions, addressing the issue of "Repeated Items Can Be Shortcuts."
- $r_3$: OOV Items Filtered Only
  - **OOV Items Filtered?** ✓
  - **Seen Items Filtered?** ✗
  - **Description**: In this case, only Out-Of-Vocabulary (OOV) items are filtered out, while all Seen Items are retained. OOV Items refer to recommendations that are not part of the predefined candidate set and may be irrelevant or invalid within the given context. By excluding OOV Items, $r_3$ ensures that only valid and recognized items are considered, thereby improving the overall quality and reliability of the recommendations.
- $r_4$: Both OOV and Seen Items Filtered
  - **OOV Items Filtered?** ✓
  - **Seen Items Filtered?** ✓
  - **Description**: This most stringent scenario filters out both OOV and Seen Items. Only new, valid, and relevant recommendations that have not been previously mentioned in the conversation are presented. By applying both filters, $r_4$ maximizes the novelty and applicability of the recommendations, ensuring alignment with the predefined candidate set and enhancing user satisfaction by avoiding redundancy.

**Summary Table**    For a concise overview, the metrics are summarized in Table 5.

**Additional Notes**

- **Out-Of-Vocabulary (OOV) Items**: These are items not included in the predefined candidate set, potentially leading to irrelevant or invalid recommendations if not filtered out.
- **Seen Items**: Items that have already been mentioned in the current conversation. Filtering these helps prevent repetitive suggestions and focuses on introducing new recommendations.

| Metric | OOV Items Filtered? | Seen Items Filtered? | Description |
|--------|---------------------|----------------------|-------------|
| $r_1$ | ✗ | ✗ | No filtering applied; includes all recommended items, serving as the upper bound for performance. |
| $r_2$ | ✗ | ✓ | Filters out Seen Items while retaining all OOV items to eliminate redundancy. |
| $r_3$ | ✓ | ✗ | Filters out OOV Items while retaining all Seen Items to ensure validity. |
| $r_4$ | ✓ | ✓ | Filters out both OOV and Seen Items, ensuring only new and valid recommendations are presented. |

Table 5: Summary of Recall@1 Metrics for Conversational Recommendation Tasks.

- **Implications for Evaluation**:
  - $r_1$: Serves as an upper bound, showcasing the model's maximum potential without any constraints.
  - $r_2$ and $r_3$: Offer intermediate evaluations, focusing on specific aspects of recommendation quality by filtering either Seen Items or OOV Items.
  - $r_4$: Provides the most stringent assessment by ensuring all recommendations are both new and valid, aligning closely with practical application scenarios.

These metrics allow us to comprehensively evaluate the effectiveness of our recommendation system under varying levels of filtering, providing deeper insights into the strengths and limitations of our approach.

## A.2 IMPLEMENTATION DETAILS

For all the experiments, we conduct experiments with eight NVIDIA-RTX-A6000 GPUs. Each experiment of SELF-PARAM needs two GPUs while all the baselines here need one GPU to run. The learning rate is set to $2 \times 10^{-5}$ for training. We train for 50 epochs in both Single Context Injection and Batch Context Injection, 20 epochs in Sequential Injection, and 1 epoch in Conversational Recommendation. The KL divergence is computed using the `torch.nn.functional.kl_div` function from the PyTorch library. For the backbone model Openllama-3B-v2, we train the MLP layers. For Mistral-7B, Mistral-7B-instruct-v0.2, Llama3-8B, we use LoRA (Hu et al., 2021) from the package `peft` (Mangrulkar et al., 2022). The LoRA configurations are:
```
{inference_mode: false, r: 8, lora_alpha: 32, lora_dropout:
0.1, target_modules: ["q_proj", "v_proj", "k_proj", "up_proj",
"down_proj", "gate_proj"] }.
```

## A.3 CONSTRUCTION OF TARGET SENTENCE SET

The prompt used for querying the instruct model using the context is shown below:

```
Given a context, please generate related questions as
comprehensively as possible with bullet points and answers.
This is an example:
Context:  A small coastal town has a beach known for its colorful
sea glass.  The town hosts an annual festival celebrating this
unique feature with art and conservation efforts.
Question:  What attracts tourists to the small coastal town
annually?  Answer:  The unique sea glass beach.
Question:  What is celebrated at the town's annual festival?
Answer:  The natural phenomenon of sea glass.
Question:  What type of activities are featured at the festival?
```

| Model & # of Contexts | 50 QA pairs (1 Epoch) | 10 QA pairs (5 epochs) |
|---|---|---|
| OpenLlama-3B-v2 & 100 | 0.5082 | 0.4203 |
| OpenLlama-3B-v2 & 500 | 0.5048 | 0.4203 |
| Mistral-7B & 100 | 0.4521 | 0.3891 |
| Mistral-7B & 500 | 0.4384 | 0.3813 |

Table 6: Sensitivity to Target Sentence Set Diversity. Here we consider batch context injection.

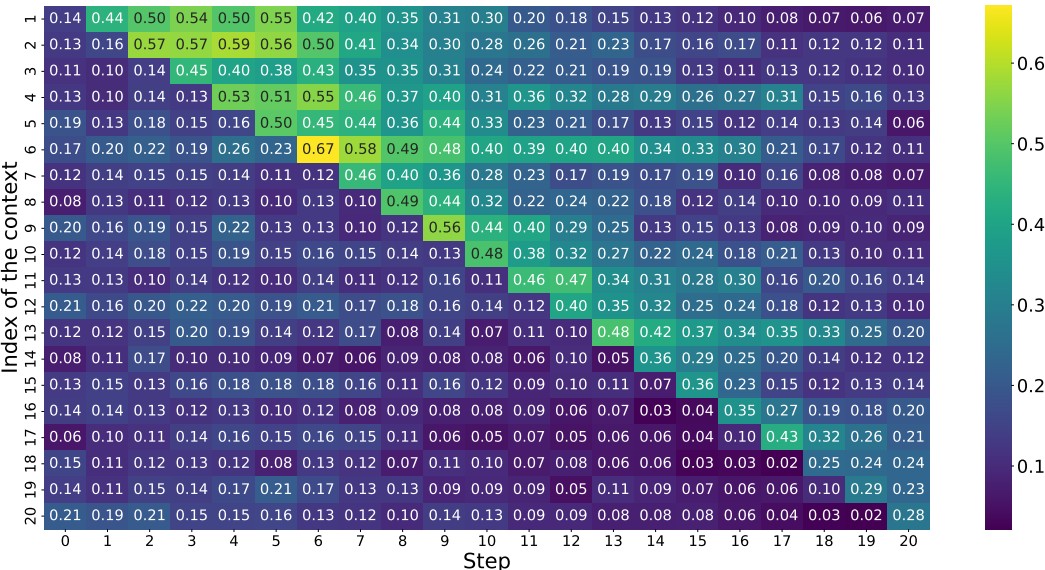

Figure 3: Average QA-F1 scores after sequentially injecting contexts into the model across 50 sequences without SlimPajama

```
Answer:  Glass art exhibitions, environmental workshops, and local
music performances.
Question:  What is the purpose of the workshops at the festival?
Answer:  To promote environmental awareness among visitors.
Question:  How does the festival impact the local economy?
Answer:  It boosts local businesses by attracting tourists.
Now, please generate related questions based on the following
context:
Context:  {context}
```

# B  ADDITIONAL EXPERIMENTS

## B.1  SENSITIVITY TO TARGET SENTENCE SET DIVERSITY

Our training objective, Eq.(3), uses 50 question-answer pairs as the target sentence set per context over one epoch. During our exploration of this project, we found that training on a smaller set of 10 question-answer pairs for five epochs led to lower performance. We report the results we obtained in Table 6. This analysis suggests that question-answer diversity is important to ensure comprehensive attention to context information. Limited diversity may lead to slightly reduced performance.

## B.2  ABLATION STUDY FOR SEQUENTIAL INJECTION

To conduct the ablation study for sequential injection, we consider the following settings:

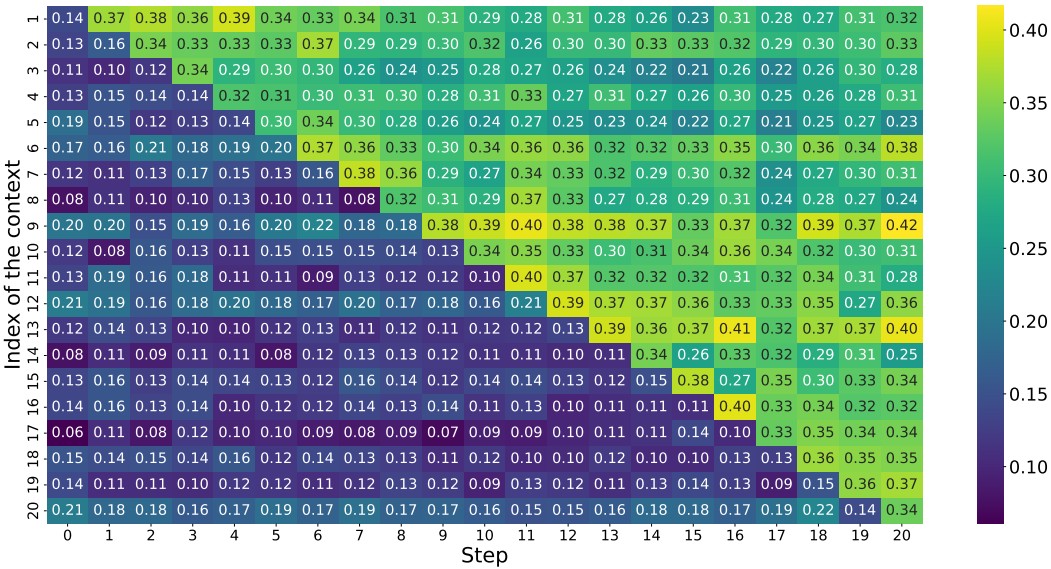

Figure 4: Average QA-F1 scores after sequentially injecting contexts into the model across 50 sequences with fine-tuning on the Target Sentence Set.

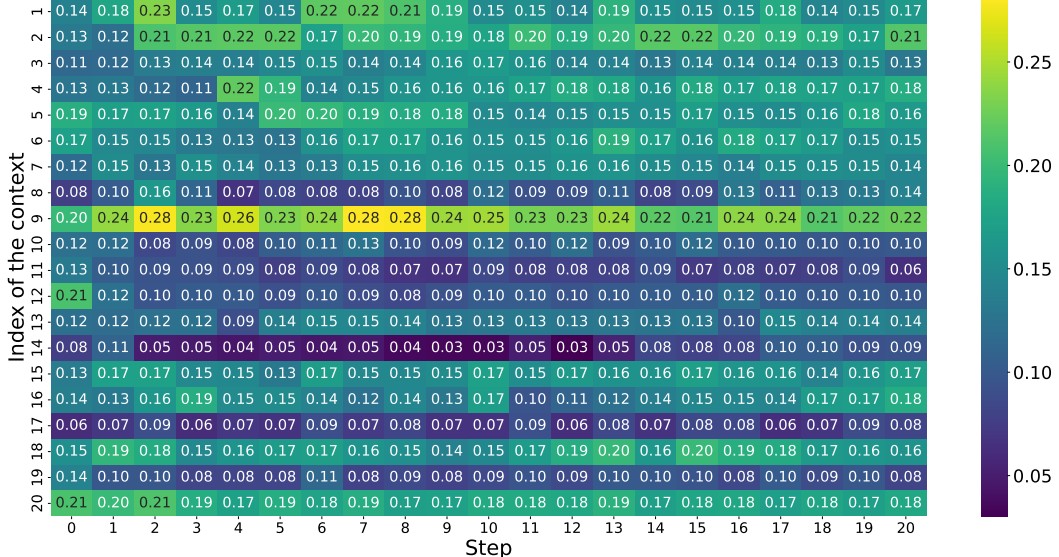

Figure 5: Average QA-F1 scores after sequentially injecting contexts into the model across 50 sequences with fine-tuning on the context.

- **SELF-PARAM w/o SlimPajama**: As mentioned in Section A.3, we adopt SlimPajama as the training set to retain the model's general abilities. Thus we conduct the experiments with exactly the same settings while skipping the steps of training on SlimPajama. The results are shown in Figure 3. Looking at the dianogal results [0.43, 0.57, 0.44, 0.52, 0.5, 0.67, 0.45, 0.49, 0.55, 0.48, 0.46, 0.4, 0.48, 0.35, 0.36, 0.34, 0.42, 0.24, 0.28, 0.27], we can see that the model's generalization ability has be severely affected after around 20 steps of injection, where the model can only achieve less than 0.30 QA-F1-score when we are using Eq.(2) to inject new knowledge. In contrast, as shown in Figure 2, SELF-PARAM can maintain the QA-F1-score as always around 0.5. This justifies that we need SlimPajama to maintain the model's general functionality.

- **Finetuning on the Context**: Following the settings FT (C) in Section 4.1 and 4.2, at each step, we fine-tune the base model, Mistral-7B, with LoRA applied, on a single context for 50 epochs. Similarly, SlimPajama is used as the regularization set during fine-tuning. After completing each step, we merge the LoRA weights into the original model before proceeding to fine-tune on the next context. To evaluate performance, we construct 25 sequences, each consisting of 20 consecutive contexts, and report the average performance across all 25 sequences (These settings are consistent with those described in Section 4.3). The results are shown in Figure 4. From the figure, we can see that FT (C) demonstrates the lowest efficacy. By omitting the first column and examining the diagonal values as highlighted, it is evident that FT (C) barely injects knowledge into the model.

- **Finetuning on the Target Sentence Set**: Following the settings FT (S) in Section 4.1 and 4.2, we fine-tune the base model on the constructed question-answering pairs (i.e., the target sentence set) constructed from `gpt-4o-mini` with the same experimental settings as SELF-PARAM, with the results shown in Figure 5. From the figure, we can find that FT (S) has a retention ability comparable to ours. As shown in the first row of Figure 2 and 5, which shows the accuracy of querying the model with the question related $x_1$ after injecting $x_1, \cdots, x_n, n = 1, \cdots, 20$ into the model (here $x_1, \cdots, x_{20}$ refer to 20 contexts that are sequentially injected), Our approach starts at a QA-F1 score of around 0.5 after injecting x1 and drops to approximately 0.33 after injecting x20 (Figure 2). In contrast, FT (S) begins at 0.36 and decreases to 0.31. Despite the similar retention ability, FT (S) has much lower efficacy, as shown by the diagonal values of the figures (meaning the QA-F1 score when answering the question related to $x_n$ after injecting $x_1, \cdots, x_n$ into the model).

