# OpenReview forum: "Self-Updatable Large Language Models by Integrating Context into Model Parameters"
_ICLR.cc/2025/Conference — ICLR 2025 Poster_

### Official Review · Reviewer_s4VY · 2024-10-31

**Soundness:** 2
**Presentation:** 3
**Contribution:** 2
**Rating:** 5
**Confidence:** 3

**Summary:**

The paper presents a context injection method SELF-PARAM.
Given a context, SELM-PARAM (i) employs an instruct model to generate a set of contextually relevant question-answering pairs (ii) trains the model by minimizing the KL divergence with the distribution where the context is provided.
Empirical evaluation is conducted to inject the contexts in PwC dataset into OpenLlama-3B-v2, Mistral-7B, and Llama-3-8B and INSPIRED and REDIAL into Mistral-7B, where the proposed method achieves consistently superior performance than the baselines in various settings.

**Strengths:**

* The proposed method requires no additional parameter and storage, which is computationally and memory efficient and compatible with the standard serving engine.
* In the empirical evaluation, the proposed method achieves superior performance in various settings, which seems inspiring.

**Weaknesses:**

* My primary concern lies in the novelty.
Prompting highly capable language models have been a common method to collect training data and KL divergence is a common loss in knowledge distillation.
* Eq. (2) is ill-posed.
The summation is over any sentence $s$, making the term infinite and cannot be minimized.
It is also unclear to me what is the KL divergence between two scalar rather than distributions.
* Unrelated sentences are randomly sampled from SlimPajama to maintain the linguistic capabilities, but relevant ablation and evaluation are missing.

**Questions:**

* Why the fine-tuning is not compared in the sequential injection setting?

---

> ### Author Response · Authors · 2024-11-15
> **Rebuttal to Reviewer s4VY**
>
> We thank the reviewer for the comments, where we address as below:
>
> **[W1] Novelty of the Approach**:  We would like to emphasize that SELF-PARAM is not merely a case of "collecting training data from highly capable models" or "using KL divergence," as this characterization overlooks the unique structure of our approach.
> - **Distinct from Data Collection and Fine-Tuning:** Although `gpt-4o-mini` is more advanced, the question-answer pairs could also be constructed using models such as `mistral-7B-instruct`. We have conducted a quick experiment using `mistral-7B` as the base model (i.e., Column Mistral-7B in Table 2) and generate 50 question-answering pairs using `mistral-7B-instruct-v0.3` and train `Mistral-7B` using these question-answering pairs with our proposed method SELF-PARAM. The experimental results are shown below:
>
>     |  | Injecting 100 Contexts into `mistral-7B` |
>     | ---- | ---- |
>     | `gpt-4o-mini` | 0.4521 |
>     | `mistral-7B-instruct-v0.3` | 0.4502 |
>
>     From the table we can see that changing the model from `gpt-4o-mini` to `mistral-7B-instruct` does not affect the performance, showing that this process does not rely on the inherent superiority of the external model but rather that the Target Sentence Set needs to be diverse. We will add the experiments on 500 contexts and `Llama3-8B-Instruct` for `Llama3-8B` in our paper as the ablation study. Moreover, The role of the question-answer pairs is more akin to preprocessing the context as all the question-answering pairs are from the context and they essentially do not have anything new other than the context itself, making it different from collecting an entirely new dataset.
>
>
> - **Baselines Demonstrating the Need for Our Approach:** Simply using another model to construct question-answering pairs from the context and fine-tuning the model does not yield good performance, as demonstrated by the baselines `FT (S), Q` in Tables 1 and 2, `FT (Q)` in Table 3.
>
>     In these baselines:
>     - **`FT (S), Q` and `FT (Q)`**: Here, the model is fine-tuned on question-answer pairs generated by a more advanced model. However, without our KL-divergence-based objective, this approach struggles to match the efficacy of SELF-PARAM. The significant performance gap shows that the mere inclusion of data from a larger model does not inherently improve long-term retention or context integration.
>
>     Together, these baselines demonstrate that collecting data from a larger model alone, even with fine-tuning, is insufficient for the nuanced task of context injection. SELF-PARAM leverages KL divergence in a way that aligns model outputs across contexts, directly embedding context within model parameters.
>
> - **Innovative Use of KL Divergence:** Although KL divergence has been introduced long ago, prior works such as parameter-based injection of prompts or factual statements [1, 2] show that this technique can still underpin novel approaches to knowledge embedding. SELF-PARAM extends this idea to address the challenge of context injection where the context can be of more complicated forms, leveraging KL divergence in a way that facilitates both long-term retention and efficient context integration without additional parameters.
>
> [1] Prompt Injection: Parameterization of Fixed Inputs
> [2] Propagating Knowledge Updates to LMs Through Distillation
>
> **[W2] Clarification of Equation and KL Divergence**: To address concerns about the formulation in Eq. (2), we plan to revise the equation as follows:
>
> $$\hat{\theta} = \arg \min_{\hat{\theta}} KL\left[p_\theta (\cdot | x) \big|\big| p_{\hat{\theta}}(\cdot)\right]$$
>
> In this form, the KL divergence is computed over the output distribution conditioned on input $x$. This requires summing over sentences in the distribution $p_\theta(\cdot|x)$, which necessitates constructing a Target Sentence Set. We will update the paper with this equation and add corresponding clarifications.
>
> As for [W3], regarding the ablation of SlimPajama, we are running a sequential injection baseline without using SlimPajama to regularize the model. As the sequential injection takes around two days to run, we will post the results a little later.
>
> As for the question [Q1], we provide the following response:
> [Q1] Comparison of Fine-Tuning in Sequential Injection Setting**: Given that fine-tuning did not achieve high efficacy in single-context and batch-context injection settings, we deemed it unnecessary to test its retention ability in the sequential injection setting, as it does not meet the required efficacy baseline. However, to provide clarity, we conducted additional fine-tuning experiments to explicitly show the difference between this baseline and our method. As the sequential injection takes some time to run, we will make a new comment below later about these results.

---

> ### Author Response · Authors · 2024-11-16
> **Experiments of [W3]**
>
> **[W3] Ablation Study on SlimPajama Usage**: As an ablation for Section 4.3 Sequential Injection, we remove the use of SlimPajama dataset during training and show the sequential injection results as below:
>
> |    0 |    1 |    2 |    3 |    4 |    5 |    6 |    7 |    8 |    9 |   10 |   11 |   12 |   13 |   14 |   15 |   16 |   17 |   18 |   19 |   20 |
> |-----:|-----:|-----:|-----:|-----:|-----:|-----:|-----:|-----:|-----:|-----:|-----:|-----:|-----:|-----:|-----:|-----:|-----:|-----:|-----:|-----:|
> |0.13|**0.43**|0.49|0.54|0.49|0.54|0.41|0.39|0.35|0.30|0.30|0.20|0.18|0.15|0.13|0.12|0.09|0.07|0.06|0.06|0.07|
> |0.13|0.16|**0.57**|0.56|0.58|0.56|0.49|0.40|0.34|0.30|0.27|0.25|0.21|0.23|0.17|0.15|0.16|0.11|0.12|0.12|0.10|
> |0.11|0.10|0.13|**0.44**|0.39|0.38|0.43|0.34|0.35|0.30|0.24|0.21|0.20|0.19|0.18|0.13|0.10|0.12|0.12|0.11|0.10|
> |0.12|0.10|0.14|0.13|**0.52**|0.51|0.55|0.45|0.36|0.39|0.30|0.35|0.31|0.28|0.28|0.25|0.26|0.30|0.14|0.16|0.12|
> |0.18|0.12|0.17|0.14|0.16|**0.50**|0.45|0.43|0.35|0.43|0.33|0.22|0.21|0.17|0.13|0.14|0.12|0.13|0.13|0.13|0.06|
> |0.17|0.20|0.21|0.18|0.25|0.23|**0.67**|0.57|0.49|0.48|0.40|0.39|0.40|0.40|0.34|0.33|0.30|0.20|0.16|0.12|0.11|
> |0.11|0.14|0.14|0.14|0.13|0.11|0.11|**0.45**|0.39|0.36|0.28|0.22|0.17|0.18|0.17|0.18|0.09|0.15|0.08|0.07|0.06|
> |0.07|0.12|0.10|0.11|0.12|0.09|0.12|0.09|**0.49**|0.43|0.32|0.22|0.23|0.22|0.18|0.11|0.14|0.09|0.09|0.09|0.11|
> |0.19|0.16|0.19|0.14|0.22|0.13|0.13|0.09|0.12|**0.55**|0.43|0.40|0.29|0.24|0.13|0.14|0.12|0.08|0.09|0.09|0.08|
> |0.12|0.13|0.17|0.14|0.18|0.14|0.16|0.15|0.13|0.12|**0.48**|0.37|0.32|0.27|0.22|0.23|0.18|0.20|0.13|0.10|0.10|
> |0.13|0.12|0.10|0.13|0.11|0.10|0.14|0.10|0.11|0.15|0.11|**0.46**|0.47|0.33|0.30|0.28|0.30|0.15|0.19|0.15|0.13|
> |0.20|0.16|0.19|0.21|0.20|0.18|0.20|0.16|0.18|0.15|0.14|0.12|**0.40**|0.35|0.32|0.25|0.24|0.18|0.12|0.13|0.09|
> |0.12|0.11|0.14|0.20|0.18|0.14|0.12|0.17|0.07|0.14|0.07|0.10|0.09|**0.48**|0.41|0.37|0.33|0.35|0.32|0.24|0.19|
> |0.07|0.11|0.17|0.10|0.10|0.09|0.06|0.05|0.08|0.07|0.08|0.06|0.10|0.05|**0.35**|0.29|0.24|0.19|0.13|0.11|0.11|
> |0.13|0.14|0.13|0.16|0.17|0.18|0.17|0.15|0.11|0.15|0.12|0.08|0.10|0.10|0.07|**0.36**|0.23|0.15|0.11|0.13|0.13|
> |0.13|0.14|0.13|0.11|0.12|0.10|0.11|0.08|0.09|0.07|0.07|0.08|0.05|0.06|0.02|0.03|**0.34**|0.27|0.19|0.18|0.20|
> |0.06|0.10|0.11|0.14|0.16|0.15|0.16|0.15|0.11|0.06|0.05|0.06|0.05|0.06|0.06|0.04|0.09|**0.42**|0.31|0.26|0.21|
> |0.15|0.10|0.12|0.12|0.12|0.08|0.12|0.11|0.07|0.10|0.09|0.06|0.07|0.06|0.05|0.02|0.03|0.02|**0.24**|0.24|0.23|
> |0.14|0.11|0.14|0.14|0.16|0.21|0.16|0.12|0.12|0.08|0.09|0.08|0.04|0.10|0.08|0.07|0.05|0.06|0.10|**0.28**|0.23|
> |0.20|0.18|0.20|0.15|0.15|0.15|0.12|0.12|0.09|0.13|0.12|0.09|0.09|0.07|0.07|0.08|0.06|0.04|0.02|0.02|**0.27**|
>
> Looking at the dianogal results `[0.43, 0.57, 0.44, 0.52, 0.5, 0.67, 0.45, 0.49, 0.55, 0.48, 0.46, 0.4, 0.48, 0.35, 0.36, 0.34, 0.42, 0.24, 0.28, 0.27]`, we can see that the model's generalization ability has be severely affected after around 20 steps of injection, where the model can only achieve less than 0.30 QA-F1-score when we are using Eq.(2) to inject new knowledge. In constrast, as shown in Figure2, our setting in the paper can maintain the QA-F1-score as always around 0.5. This justifies that we need SlimPajama to regularize the model's behavior.

---

> ### Author Response · Authors · 2024-11-16
> **Experiments of [Q1]; Part (1/2)**
>
> **[Q1] Comparison of Fine-Tuning in Sequential Injection Setting**: We have completed the experiments for the fine-tuning baseline in the Sequential Injection setting. Following the fine-tuning baselines in Table 2, we evaluate the following two specific settings:
>
> - `FT (C)`: At each step, we fine-tune the base model, Mistral-7B, with LoRA applied, on a single context for 50 epochs. SlimPajama is used as the regularization set during fine-tuning. After completing each step, we merge the LoRA weights into the original model before proceeding to fine-tune on the next context. To evaluate performance, we construct 25 sequences, each consisting of 20 consecutive contexts, and report the average performance across all 25 sequences (These settings are consistent with those described in Section 4.3). The results are summarized below.
>
>   |    0 |    1 |    2 |    3 |    4 |    5 |    6 |    7 |    8 |    9 |   10 |   11 |   12 |   13 |   14 |   15 |   16 |   17 |   18 |   19 |   20 |
>   |-----:|-----:|-----:|-----:|-----:|-----:|-----:|-----:|-----:|-----:|-----:|-----:|-----:|-----:|-----:|-----:|-----:|-----:|-----:|-----:|-----:|
>   |0.13|**0.17**|0.23|0.14|0.16|0.14|0.21|0.21|0.21|0.18|0.14|0.15|0.14|0.19|0.15|0.14|0.15|0.18|0.14|0.14|0.16|
>   |0.13|0.12|**0.21**|0.21|0.21|0.21|0.17|0.19|0.19|0.18|0.17|0.20|0.18|0.19|0.21|0.21|0.19|0.19|0.18|0.17|0.21|
>   |0.11|0.11|0.13|**0.13**|0.13|0.14|0.15|0.13|0.14|0.16|0.16|0.15|0.14|0.14|0.12|0.13|0.13|0.13|0.13|0.15|0.12|
>   |0.12|0.13|0.11|0.11|**0.21**|0.19|0.13|0.14|0.15|0.16|0.16|0.17|0.18|0.17|0.15|0.18|0.16|0.17|0.17|0.17|0.18|
>   |0.18|0.17|0.17|0.16|0.13|**0.19**|0.19|0.18|0.18|0.18|0.14|0.14|0.14|0.15|0.15|0.16|0.15|0.15|0.16|0.17|0.16|
>   |0.17|0.14|0.14|0.13|0.12|0.12|**0.15**|0.17|0.17|0.15|0.15|0.15|0.16|0.19|0.17|0.15|0.18|0.17|0.16|0.15|0.15|
>   |0.11|0.15|0.12|0.15|0.14|0.12|0.13|**0.15**|0.15|0.15|0.14|0.15|0.15|0.15|0.15|0.14|0.13|0.15|0.15|0.15|0.14|
>   |0.07|0.10|0.16|0.11|0.06|0.08|0.08|0.08|**0.09**|0.08|0.12|0.09|0.09|0.10|0.08|0.08|0.12|0.10|0.12|0.13|0.13|
>   |0.19|0.24|0.27|0.23|0.26|0.23|0.23|0.28|0.27|**0.24**|0.24|0.22|0.22|0.24|0.21|0.21|0.23|0.23|0.21|0.22|0.22|
>   |0.12|0.12|0.08|0.09|0.08|0.10|0.11|0.12|0.09|0.08|**0.11**|0.10|0.12|0.09|0.10|0.11|0.10|0.10|0.09|0.09|0.09|
>   |0.13|0.10|0.09|0.09|0.08|0.07|0.08|0.07|0.06|0.06|0.08|**0.07**|0.07|0.08|0.09|0.07|0.07|0.07|0.07|0.09|0.06|
>   |0.20|0.12|0.09|0.10|0.10|0.09|0.09|0.08|0.08|0.09|0.10|0.10|**0.09**|0.10|0.10|0.10|0.12|0.10|0.09|0.09|0.09|
>   |0.12|0.11|0.12|0.12|0.09|0.13|0.14|0.14|0.14|0.12|0.12|0.13|0.12|**0.12**|0.12|0.12|0.09|0.14|0.13|0.14|0.13|
>   |0.07|0.10|0.05|0.04|0.04|0.04|0.04|0.04|0.03|0.03|0.03|0.05|0.03|0.05|**0.08**|0.07|0.08|0.10|0.10|0.09|0.09|
>   |0.13|0.17|0.17|0.14|0.14|0.13|0.16|0.14|0.14|0.15|0.17|0.15|0.16|0.15|0.15|**0.17**|0.16|0.15|0.14|0.16|0.17|
>   |0.13|0.13|0.16|0.18|0.14|0.15|0.14|0.12|0.14|0.12|0.16|0.09|0.11|0.12|0.13|0.14|**0.14**|0.14|0.16|0.16|0.18|
>   |0.06|0.07|0.09|0.06|0.06|0.06|0.09|0.07|0.08|0.06|0.06|0.09|0.06|0.08|0.06|0.08|0.08|**0.06**|0.06|0.08|0.08|
>   |0.15|0.19|0.18|0.14|0.15|0.16|0.17|0.15|0.15|0.13|0.14|0.16|0.18|0.19|0.16|0.19|0.18|0.18|**0.16**|0.16|0.15|
>   |0.14|0.09|0.09|0.08|0.08|0.08|0.11|0.07|0.08|0.07|0.08|0.09|0.08|0.10|0.09|0.09|0.08|0.10|0.08|**0.09**|0.08|
>   |0.20|0.19|0.21|0.18|0.17|0.18|0.17|0.18|0.17|0.17|0.18|0.18|0.17|0.19|0.17|0.17|0.17|0.16|0.17|0.17|**0.17**|

---

> ### Author Response · Authors · 2024-11-16
> **Experiments of [Q1]; Part (2/2)**
>
> - `FT (S)`: This setting differs from the above one in the way that it finetunes the model on the generated question-answering pairs from `gpt-4o-mini` rather than the context. The results are shown below:
>
>   |    0 |    1 |    2 |    3 |    4 |    5 |    6 |    7 |    8 |    9 |   10 |   11 |   12 |   13 |   14 |   15 |   16 |   17 |   18 |   19 |   20 |
>   |-----:|-----:|-----:|-----:|-----:|-----:|-----:|-----:|-----:|-----:|-----:|-----:|-----:|-----:|-----:|-----:|-----:|-----:|-----:|-----:|-----:|
>   |0.13|**0.36**|**0.37**|**0.35**|**0.39**|**0.33**|**0.33**|**0.34**|**0.31**|**0.31**|**0.28**|**0.27**|**0.31**|**0.27**|**0.26**|**0.23**|**0.30**|**0.27**|**0.27**|**0.30**|**0.31**|
>   |0.13|0.16|**0.34**|0.33|0.33|0.32|0.36|0.29|0.29|0.30|0.31|0.26|0.29|0.29|0.33|0.32|0.32|0.29|0.29|0.30|0.32|
>   |0.11|0.10|0.11|**0.34**|0.29|0.30|0.29|0.26|0.24|0.24|0.27|0.26|0.26|0.23|0.21|0.20|0.26|0.21|0.25|0.29|0.28|
>   |0.12|0.15|0.14|0.13|**0.31**|0.31|0.30|0.30|0.30|0.28|0.30|0.33|0.26|0.30|0.27|0.26|0.29|0.25|0.25|0.27|0.30|
>   |0.18|0.15|0.11|0.12|0.13|**0.30**|0.33|0.30|0.28|0.25|0.24|0.26|0.24|0.23|0.23|0.21|0.27|0.21|0.24|0.26|0.22|
>   |0.17|0.16|0.20|0.18|0.19|0.19|**0.37**|0.35|0.32|0.30|0.34|0.35|0.35|0.31|0.32|0.33|0.35|0.30|0.35|0.34|0.38|
>   |0.11|0.11|0.12|0.16|0.14|0.12|0.15|**0.38**|0.36|0.28|0.26|0.34|0.33|0.31|0.28|0.30|0.32|0.23|0.27|0.29|0.30|
>   |0.07|0.11|0.09|0.10|0.13|0.09|0.10|0.07|**0.31**|0.30|0.28|0.36|0.32|0.27|0.27|0.28|0.30|0.24|0.27|0.27|0.24|
>   |0.19|0.19|0.14|0.18|0.16|0.20|0.22|0.17|0.18|**0.37**|0.38|0.40|0.38|0.37|0.37|0.33|0.36|0.31|0.39|0.37|0.41|
>   |0.12|0.08|0.15|0.12|0.11|0.14|0.14|0.14|0.14|0.13|**0.34**|0.35|0.33|0.30|0.31|0.33|0.36|0.34|0.31|0.30|0.30|
>   |0.13|0.18|0.15|0.18|0.10|0.11|0.08|0.12|0.11|0.12|0.09|**0.39**|0.36|0.32|0.32|0.31|0.31|0.32|0.34|0.30|0.27|
>   |0.20|0.19|0.15|0.18|0.19|0.18|0.16|0.19|0.16|0.17|0.15|0.21|**0.39**|0.37|0.36|0.35|0.32|0.33|0.35|0.27|0.35|
>   |0.12|0.14|0.13|0.10|0.09|0.11|0.12|0.10|0.12|0.11|0.11|0.12|0.13|**0.39**|0.36|0.36|0.40|0.31|0.36|0.36|0.40|
>   |0.07|0.10|0.09|0.11|0.11|0.08|0.11|0.13|0.12|0.12|0.11|0.11|0.10|0.10|**0.34**|0.26|0.32|0.31|0.28|0.31|0.25|
>   |0.13|0.15|0.12|0.14|0.13|0.13|0.12|0.15|0.14|0.12|0.13|0.14|0.13|0.11|0.14|**0.37**|0.27|0.34|0.30|0.32|0.34|
>   |0.13|0.15|0.13|0.14|0.10|0.11|0.12|0.13|0.12|0.14|0.11|0.12|0.09|0.10|0.11|0.10|**0.40**|0.33|0.34|0.32|0.32|
>   |0.06|0.10|0.08|0.12|0.09|0.10|0.08|0.08|0.08|0.06|0.09|0.08|0.09|0.10|0.11|0.13|0.10|**0.33**|0.34|0.34|0.34|
>   |0.15|0.14|0.15|0.13|0.15|0.11|0.13|0.13|0.12|0.10|0.11|0.10|0.10|0.12|0.10|0.10|0.12|0.12|**0.36**|0.35|0.34|
>   |0.14|0.10|0.10|0.10|0.12|0.11|0.10|0.11|0.13|0.12|0.09|0.12|0.12|0.10|0.12|0.13|0.12|0.09|0.15|**0.35**|0.36|
>   |0.20|0.17|0.18|0.16|0.16|0.18|0.17|0.19|0.17|0.17|0.16|0.14|0.15|0.16|0.18|0.18|0.16|0.19|0.22|0.14|**0.33**|
>
> From these results, we observe the following: (1) `FT (C)` demonstrates the lowest efficacy. By omitting the first column and examining the diagonal values as highlighted, it is evident that `FT (C)` barely injects knowledge into the model. (2) `FT (S)` shows a retention ability comparable to ours. As shown in the first row of the table and Figure 2, we inject `x1, x2, ..., x20` into the model and query it with a question related to x1. Our approach starts at a QA-F1 score of around 0.5 after injecting `x1` and drops to approximately 0.33 after injecting `x20` (Figure 2). In contrast, `FT (S)` begins at 0.36 and decreases to 0.31. However, `FT (S)` has much lower efficacy, as shown by the diagonal values of the table. These represent the QA-F1 score for the question related to xn after injecting xn into the model. We will incorporate these results and their analysis into our paper.

---

> ### Author Response · Authors · 2024-11-22
> **Clarification and Reconsideration**
>
> **Dear Reviewer s4VY,**
>
> Thank you again for your constructive feedback!
>
> Regarding your primary concern, we would like to clarify again that our performance gains do not stem from using a more powerful external model. In our new experiments, the QA pairs are generated using `Mistral-Instruct-7B-v0.3`, which is equally powerful as our base model, `Mistral-7B`. This demonstrates that **the effectiveness of our approach is independent of the external model’s strength**.
>
> Additionally, while KL divergence is a well-established method, existing works have not utilized it for context integration involving diverse forms of context, such as paragraphs and conversations. Our approach **uniquely leverages KL divergence to effectively integrate various types of context information into the model, distinguishing it from prior methods.**
>
> We would be extremely grateful if you could kindly reconsider our submission in light of these clarifications. Thank you once again for your time and thoughtful evaluation.
>
> Best regards,
> The Authors

---

> > ### Comment · Reviewer_s4VY · 2024-11-22
> >
> > I thank the authors for addressing my concerns.
> > I have raised my score accordingly.

---

> > > ### Author Response · Authors · 2024-11-24
> > > **Kindly Asking if the Reviewer could take some more Consideration**
> > >
> > > Dear Reviewer s4VY,
> > >
> > > Thank you for your thoughtful engagement and for acknowledging our efforts in addressing your concerns. We deeply appreciate your recognition of our work and the increase in your score (3 $\rightarrow$ 5).
> > >
> > > Given that we have resolved all your concerns and incorporated your valuable suggestions to strengthen the paper, we were wondering if it might be possible for you to kindly consider assigning a positive score, as we sincerely hope that our revisions align with the standards of ICLR and demonstrate the merit of the work.
> > >
> > > We would be extremely grateful for your reconsideration and greatly value your time and effort in reviewing our submission!
> > >
> > > Warm Regards,
> > > Authors

---

> ### Author Response · Authors · 2024-12-03
> **Follow Up about A Positive Score**
>
> Dear Reviewer s4VY,
>
> Thank you again for your thoughtful engagement and raising your score from 3 $\rightarrow$ 5! However, given that we have addressed your concerns and that 5 is still a negative score, we would like to kindly inquire whether a positive score could better reflect your evaluation to our paper. If there are more concerns regarding the paper, we would be more than happy to continue the discussion!
>
> Best,
> Authors

---

### Official Review · Reviewer_3X9B · 2024-11-02

**Soundness:** 3
**Presentation:** 3
**Contribution:** 3
**Rating:** 6
**Confidence:** 4

**Summary:**

The paper presents SELF-PARAM, a framework for enabling self-updatable large language models that integrate new knowledge directly into model parameters without additional storage modules. By minimizing the Kullback-Leibler (KL) divergence between an original model (with context access) and a target model (without context), SELF-PARAM effectively embeds context within the model’s parameters. The framework addresses the challenges of rapid experience integration and long-term retention by incorporating diverse question-answer pairs related to the injected knowledge. Experimental results show SELF-PARAM’s effectiveness across single and batch context injections, sequential knowledge injections, and conversational recommendations, outperforming baseline methods in knowledge retention and response accuracy without extra parameters or external storage.

**Strengths:**

Originality: The approach of embedding knowledge directly in model parameters through KL divergence minimization of the source & target model while excluding the context information from the target model is a creative solution to self-updateability without extra storage.
Quality: Experimental results validate the approach across multiple tasks, showing improvements over existing methods in terms of efficacy and retention.
Clarity: The methodology and experiments are largely well-explained, making the paper accessible to researchers familiar with LLMs.
Significance: Addressing rapid knowledge integration and long-term retention in LLMs is a critical step forward, particularly for applications requiring frequent updates in dynamic environments.

**Weaknesses:**

Limited Baseline Analysis: While the paper compares extensively against memory-based methods, it does not do any comparison with other methods that are similar to the proposed methods, such as regularization approaches or other distillation based approaches that are commonly used in the continual learning literature.

**Questions:**

Questions
- Could the authors elaborate on potential limitations when scaling SELF-PARAM to even larger LLMs or highly complex contexts?
- How sensitive is the model’s performance to variations in the diversity of the question-answer pairs generated for context injection?
Suggestions
- I would suggest that the authors add other baselines that do not add additional modules or parameters; they are mentioned in the introduction section, but no explicit comparisons were made in the experimental section. Since this paper highlights the component that no additional parameters are needed, such comparisons should be made with those methods as well.

---

> ### Author Response · Authors · 2024-11-15
> **Rebuttal to Reviewer 3X9B**
>
> We sincerely thank the reviewer for the insightful comments! We address the concerns as below:
>
> **[W1] Baseline Analysis and Use of Regularization**: We appreciate the reviewer’s feedback and will clarify in the paper that we incorporate regularization to mitigate catastrophic forgetting by fine-tuning on Slimpajama data while training with contexts or questions, shown as `FT (C), Q` and `FT (S), Q` in Tables 1 and 2. This regularization approach, commonly used in continual learning (e.g., retention sets or replay buffers), helps maintain prior knowledge, as seen in related work [1, 2]. Regarding distillation-based methods, while they typically target prompt or factual statement integration, they are not directly applicable for our goal of embedding more complex contexts, such as paragraphs or conversations, directly into the model. We will emphasize the use of Slimpajama as a retention mechanism in the paper to clarify this aspect of our approach.
>
> **[Q1] Limitations of Scaling SELF-PARAM to Larger LLMs or Complex Contexts**: We acknowledge the challenge of scaling SELF-PARAM to larger LLMs, where full fine-tuning may become impractical. Low-Rank Adaptation (LoRA) can address this limitation by enabling more efficient parameter updates; however, LoRA may slow convergence, which could impact training efficiency. For highly complex contexts, another potential limitation arises when `gpt-4o-mini` may be unable to generate question-answer pairs that fully capture the context’s nuances, potentially leading to underutilized information during KL-divergence training. To mitigate this, we could explore generating a more diverse question-answer set and filtering redundant pairs to better preserve the depth of complex contexts during training.
>
> **[Q2] Sensitivity to Question-Answer Diversity in Context Injection**: Our training objective, Eq. (2), uses 50 question-answer pairs per context over one epoch. During our exploration of this project, we found that training on a smaller set of 10 question-answer pairs for five epochs led to lower performance. We report the results we obtained below:
>
> | Model & # of Contexts| 50 QA pairs (1 Epoch) | 10 QA pairs (5 Epochs) |
>   | -- | -- | -- |
>   | OpenLlama-3B-v2 - 100 | 0.5082 | 0.4203 |
>   | OpenLlama-3B-v2 - 500 | 0.5048 | 0.4203 |
>   | Mistral-7B - 100 | 0.4521 | 0.3891 |
>   | Mistral-7B - 500 | 0.4384 | 0.3813 |
>
> This analysis suggests that question-answer diversity is important to ensure comprehensive attention to context information. Limited diversity may lead to slightly reduced performance. We will add this analysis and these results into our paper.
>
> **[Q3] Comparison with Parameter-Free Methods**: In our related work section, we referenced three categories of methods without additional parameters: (1) Continual Learning, (2) Model Editing, and (3) Knowledge Distillation. As noted in [W1], the baselines `FT (C), Q` and `FT (S), Q` in Tables 1 and 2 apply regularization-based continual learning (category (1)). However, methods in categories (2) and (3) are typically restricted to embedding factual statements or prompts, making them not directly applicable in our setting where we focus on integrating broader contexts like paragraphs or conversations. We would like to emphasize that methods without additional parameters are indeed compared in our experiments, fulfilling the scope of parameter-free comparisons.
>
> [1] Improving Information Retention in Large Scale Online Continual Learning
> [2] Active Continual Learning: on Balancing Knowledge Retention and Learnability

---

> ### Author Response · Authors · 2024-11-22
> **Clarification and Reconsideration**
>
> Dear Reviewer 3X9B,
>
> We sincerely thank you for your constructive feedback and valuable suggestions.
>
> Regarding the major weakness you mentioned, we would like to clarify again that our continual learning baselines, `FT (Q)` and `FT (C)`, are indeed implemented as **continual learning baselines with regularization** from SlimPajama. Then **knowledge distillation** for context integration is underexplored. Though previous methods are related, they are not directly applicable for context integration where the context can be of various forms (from paragraphs to conversations). The limitation of existing knowledge distillation works  also underscores the necessity of our method, and we do acknowledge that our method is inspired by the existing knowledge distillation paradigm.
>
> We would be extremely grateful if you could kindly reconsider our submission in light of these clarifications. Thank you once again for your time and thoughtful suggestions.
>
> Best regards,
> The Authors

---

> > ### Comment · Reviewer_3X9B · 2024-11-25
> > **Thank you for the rebuttal**
> >
> > The authors have relieved most of my concerns, thus I have raised my score from 5 -> 6.

---

> ### Author Response · Authors · 2024-11-25
> **Thank you for your appreciation and recognition**
>
> Dear Reviewer 3X9B,
>
> We sincerely appreciate your thoughtful feedback and the recognition of our work. Thank you for your time and effort in reviewing our submission!
>
> Best wishes,
> Authors

---

### Official Review · Reviewer_CZiX · 2024-11-03

**Soundness:** 2
**Presentation:** 3
**Contribution:** 3
**Rating:** 6
**Confidence:** 3

**Summary:**

The authors propose a method to inject context information into LLMs on-the-fly, with two criteria: being able to effectively use the injected the context (efficacy), and retain context information for a long time, as more additional information is added (retention).
The method involves two key steps:

1. An external LLM is used to augment the given context information by generating relevant question-answer pairs.
2. The base LLM is fine-tuned on the question-answer pairs, using KL-divergence.
Notably, the base LLM is fine-tuned such that $P(A_{i} | Q, A_{0, ..., i-1})$ follows $P(A_{i} | C, Q, A_{0, ..., i-1})$. I.e., the LLM is trained to answer the plain question (where the context had *not been* given), as if the context *had been* given.

The authors evaluate the method under three settings:

1. Single context injection: inject a single piece of context information and evaluate on a held-out question.
2. Batch context injection: inject a batch (100 to 500 pieces) of context information and evaluate on held-out questions.
3. Sequential injection: inject contexts 0 to i and evaluate on question for context i (for i in 0 ... 20).

The authors compare with the following baselines:

1. prompting with the relevant context (upper-bound)
2. fine-tuning on the context itself
3. LLMs with external memory
4. infinite-context methods (InfLLM)
5. RAG methods

The proposed method outperforms all previous methods (2–5), often by a wide margin.

**Strengths:**

1. The method substantially outperforms previous methods.
1. The method is empirically validated on three evaluation settings, across three base LLMs, comparing with many baseline methods.
1. The writing is easy to follow and the experiment results are presented clearly.

**Weaknesses:**

1. **The requirement of an external teacher LLM is heavily underemphasized**: Throughout the paper, the authors emphasize that the main component of the proposed method is the *KL divergence objective* applied to the base model. However, the augmented context information (QA pairs) generated by the *external model* appear to play a significant role and come with non-trivial costs. This highlights several points of potential improvement:

    1. **Cost implications of using an external teacher LLM**: The external LLM used to augment context information (gpt-4o-mini) is considerably more advanced than the base models considered in this study (OpenLLaMA 3B, Mistral 7B, Llama 3 8B), effectively positioning it as a "teacher" model. This substantial cost factor is neither discussed nor mentioned in the paper and could benefit from additional clarity.

    1. **Unclear role of KL divergence in observed performance gains**: Given the role of augmented context information (QA pairs) generated by the external LLM, it remains uncertain how much of the observed performance gains stem directly from the KL divergence objective. A useful ablation study might involve comparing the proposed method with a fine-tuning baseline that employs next-word prediction (NWP) on the generated QA samples, rather than KL divergence with the context-prompted base LLM. This could clarify the unique contribution of the KL divergence. Note that the fine-tuning baseline experiment in the paper only uses the context information itself, rather than the augmented QA pairs.

    1. **Potentially misleading title**: Given the dependency on an external LLM, it may be misleading to describe the method as enabling "self-updatable LLMs". Additionally, describing fine-tuning (using KL divergence) as “updatable LLMs with parameter integration” may over-exaggerate the scope of the work.

1. **The novelty of the proposed context injection method may be overstated**: As described in Lines 124-128, previous works have also introduced the approach of distilling a given prompt into the model by replicating the output distribution when the prompt has been prepended (e.g., Choi et al. 2022), extending to distilling factual knowledge (context information) as well (Padmanabhan et al. 2024). Claims such as “This paper introduces the concept of Context Injection in LMs” (Line 154) might be reconsidered for a more balanced presentation.

1. **Lack of discussion on the costs associated with fine-tuning**: The computational cost of fine-tuning LLMs is substantially higher than that of inference, typically requiring roughly triple the computation due to backward passes, and significantly more memory to store intermediate activations. This additional cost is unique to fine-tuning approaches and does not apply to non-fine-tuning alternatives (e.g., MemoryLLM, InfLLM, RAG methods). However, the paper only contrasts storage requirements, which might downplay the runtime costs associated with fine-tuning in the proposed method.
    1. **Insufficient clarification of storage requirements across methods**: Providing more details on the storage requirements of the baseline methods—specifically by showing the constant factor (e.g., number of bytes) relative to the full model size—could be beneficial to readers, alongside asymptotic complexity.
    1. **Limited applicability of the method for "rapid integration"**: While the paper discusses the need for “rapid and frequent integration of small-scale experiences” (Lines 11-12), it is unclear how the fine-tuning-based approach proposed here supports rapid integration. A clarification might help strengthen this point.
        - To achieve rapid integration in a ChatGPT-like setting, one could use the context window to retain current session information and employ the proposed method (or baseline methods) to store and retrieve context from prior sessions. Fine-tuning could occur between sessions to support this integration. This may represent a more realistic evaluation setting. This may also achieve higher overall efficacy, as evidenced by the results in the single-context injection scenario, where the base model with the context provided directly in the prompt outperforms all other methods.

1. **Potential risk of encouraging hallucination**: Standard LLM pretraining typically encourages the model to respond accurately to the given context. Under the proposed method, however, the model is fine-tuned to respond to a question \( Q \) as though a specific context \( C \) were provided, even when it is not. This could introduce a risk of hallucination (and other undesirable behavior) by training the model to produce responses that might not be contextually relevant. For example, if the model is trained with QA pairs based on the context “Assume the current year is 1985,” it might answer a question like “Who is the current president?” with “Ronald Reagan,” disregarding the actual context. *(This is an illustrative example to convey the potential issue.)*

*This section was written by the reviewer, re-worded using ChatGPT, and manually checked by the reviewer*.

**Questions:**

Please refer to weaknesses.

---

> ### Author Response · Authors · 2024-11-15
> **Rebuttal to Reviewer CZiX (Part 1/2)**
>
> We sincerely thank the reviewer for all these constructive points! We provide the following detailed responses to each point:
>
> **[W1-1] Cost implications of using an external teacher LLM**: We appreciate the reviewer’s emphasis on this point. We anticipated the potential impact of the information augmented by an advanced model, so we included an experiment that fine-tunes on the Target Sentence Set (QA pairs generated by `gpt-4o-mini`), denoted as `FT (S), Q` in Tables 1-2 and `FT (Q)` in Table 3. As mentioned in Lines 358–360, “Meanwhile, FT (S) performs better than FT (C) but still falls short of SELF-PARAM, highlighting that our method does more than merely injecting generated QA pairs—it facilitates the model’s ability to internalize and dynamically apply the context.” Thus, while the Target Sentence Set may potentially introduce knowledge from the external model, the primary gain stems from the design of Eq. (2) and the KL-divergence training. We will further clarify this point in our experiments to demonstrate the effectiveness of our framework.
>
> **[W1-2] Role of KL Divergence in Observed Performance Gains**: As indicated in [W1-1], we argue that our method’s performance advantage primarily arises from the training structure we introduce, rather than the Target Sentence Set or external model knowledge alone. To make this distinction clearer, we will add further clarifications on the role of `FT (S), Q` as a baseline in our experiments.
>
> **[W1-3] Clarification on the Title**: We understand the reviewer’s concern and will adjust our language accordingly.
>   - By “self-updatable,” we refer to the model’s capacity to integrate new data autonomously. While we used gpt-4o-mini to generate the Target Sentence Set in our experiments, this task could be handled by the model itself, given sufficient capacity (e.g., mistral-7b-instruct). We conduct the experiments on `mistral-7B` with batch context injection (the same setting as the column `Mistral-7B` in Table 2), while instead of using `gpt-4o-mini` to generate the question-answering pairs, we use `Mistral-7B-Instruct-v0.3` to run the generation, obtaining 50 question-answering pairs for each context. Then we run the experiments of batch context injection with 100 contexts and show the results below:
>     |  | 100 Contexts |
>     | ---- | ---- |
>     | `gpt-4o-mini` | 0.4521 |
>     | `mistral-7B-instruct-v0.3` | 0.4502 |
>
>     From this table we can see that our method is still effective even without using a more advanced model to construct the question-answering pairs. Moreover, In practical applications, preprocessing via APIs (such as gpt-4o-mini) can be used to handle incoming data before self-updating, which serves as preprocessing rather than altering the model’s inherent updating mechanism. We will clarify this terminology to avoid any potential ambiguity in our description.
>   - By "Updatable LLMs with Parameter Integration", we aim to demonstrate that we can update the parameters to integrate the context into the model parameters. To avoid exaggeration, we propose to change our title to "Self-Updatable Large Language Models with Integrating Context into Model Parameters".
>
> **[W2] Novelty and Terminology Clarification for Context Injection**: We appreciate the reviewer’s insights on distinguishing our approach from related work. While earlier methods have proposed integrating prompts or factual statements, our concept of “context injection” specifically addresses integrating various context types (e.g., paragraphs or conversations) into the model. We will adjust our phrasing to “We focus on the context injection task, defined as ...” instead of “We introduce” to ensure our claims are presented with appropriate nuance.

---

> ### Author Response · Authors · 2024-11-15
> **Rebuttal to Reviewer CZiX (Part 2/2)**
>
> **[W3-1] Storage Complexity Across Categories**: Our goal with Storage Complexity was to represent the characteristic complexity within each category of methods, inspired by [1]. While we chose a set of representative baselines, we intended this analysis to convey that other methods within each category exhibit similar Storage Complexity. By highlighting category-level complexity rather than specific bytes, we aim to offer a more generalized perspective. We will refine our wording to clarify our intentions.
>
> **[W3-2] Applicability to Rapid Integration and Session-Based Scenarios**: We thank the reviewer for this insightful suggestion. Based on further research, we found that LOCOMO [2] could be adapted to our setting, possibly by dividing interactions into session-like segments based on date, even if they lack a formal session structure. We find this a promising direction and intend to include it in our future work.
>
> **[W4] Hallucination Mitigation** We appreciate the reviewer’s example of “Assume the current year is 1985” as it effectively highlights the potential risk of hallucination. To mitigate such issues, we filtered out prompts that require direct access to prior context in the original PwC dataset. For example, prompts such as “summarize the previous context” and “write a title for the above text” were removed, leaving only factual questions. This refined dataset contains context and questions that are based on constant knowledge, as illustrated below:
>
>     - Question: What language was spoken in the tape recording of Jamal Khashoggi's murder? Answer: Arabic
>     - Question: Identify the start date for United Airlines' service from Paine Field in Everett, Washington. Answer: March 31
>
> These factual statements are less likely to induce hallucination, which helps support the method’s robustness.
>
> Regarding the Target Sentence Set, we are not directly fine-tuning the model to predict each word within this set. Instead, we employ KL divergence to transfer the behavior from the original model to the updated one, which may reduce the risk of hallucination. We appreciate the reviewer’s valuable feedback and will further review the Target Sentence Set to ensure that any problematic cases are addressed. Removing examples that may induce hallucination is unlikely to reduce our method’s efficacy, as such examples would typically increase the KL divergence, making optimization more challenging.
>
>
> [1] Towards LifeSpan Cognitive Systems.
> [2] Evaluating Very Long-Term Conversational Memory of LLM Agents

---

> ### Comment · Reviewer_CZiX · 2024-11-18
> **Thank you for addressing my concerns**
>
> **[W1] Requirement of teacher LM: largely resolved.**
>
> I'm apologize  for overlooking the FT (S) experiment. It seems that the proposed method significantly outperforms this baseline. I appreciate the results using mistral 7B for augmentation, and I believe it significantly supports the self-updatability aspect. Here are some suggestions for the authors.
>
> - Clearly identify and acknowledge the role of both (1) context augmentation and (2) KL divergence loss in the proposed method. While self-param significantly outperforms FT (S), there is still a significant gap between Base Q, FT (C) <--> FT (S). This is not acknowledged in the paper,  and does not seem to be discussed adequately in the paper, e.g., the discussion in Section 3.3 only mentions FT (C).
> - Expand and emphasize the results using a non-external model (e.g. mistral 7b) for context augmentation, with results in the appendix, and preferably the main paper.
>
> **Remaining question**: can the authors please confirm that identical experimental settings were used for the proposed method and fine-tune (S)? I.e., same generated QA pairs, same epochs, learning rate, or same level of hyperparameter optimization. I will revise my score if the authors confirm this.
>
> .
>
> **[W1-1] Cost implications of inference to augment context with QA pairs and KL fine-tuning: concerns still stand**.
>
> While the method does not require a *more capable* external model, there is still significant cost remaining in QA pair augmentation and fine-tuning.
>
> - I strongly disagree with the notion that the use of gpt4o-mini `serves as preprocessing rather than altering the model’s inherent updating mechanism.` The fact that gpt4o-mini is provided as a seemingly inexpensive API does not mean we can ignore its cost–the requirement of gpt4o-mini means that the method requires additional inference using an external model.
>   - *If* the method did not work using mistral 7B for QA pair augmentation and did in fact require a more capable model, then (1) it would incur significant runtime costs, (2) and we could not consider this a "self-updatable" method, and thus e.g. it could not be applied to the most superior frontier models. This concern has been resolved by the mistral 7B results.
> - Runtime cost: compared to using the base model, the proposed method requires significant inference and fine-tuning costs. Where the base model would have simply needed to process (prefill) the context at the time it was given, the proposed method requires the external model to generate numerous question answer pairs for each context and KL fine-tune the original model on that data. Let's say that the base model has a inference cost equivalent to \\$1 to process some context; generating single QA pair for each context using the external model is \\$5; KL fine-tuning the data on the base model is \\$5. Then, for every piece of context, that would have costed \\$1 to be initially processed by the base model, the proposed method would require an extra \\$10 to "internalize" that context, *for each QA pair generated*. I.e., a ChatGPT-like service equipped with the given method for context-preservation (beyond the attention window) would cost 10x to deploy.
>   - **Question: how many QA pairs were generated per context?**
>
> I urge the authors to acknowledge and adequately discuss these costs in the paper.
>
> .
>
> **[W2] Novelty and Terminology Clarification for Context Injection: resolved.**
>
> **[W3-2] Rapid integration: concerns still stand**. I suggest the authors to discuss the limitations of the proposed method in rapid integration, noting that it can be mitigated with approaches such as LOCOMO.
>
> **[W4] Hallucination: resolved.** I appreciate the attention to detail regarding this issue. However regarding `"we employ KL divergence to transfer the behavior from the original model to the updated one, which may reduce the risk of hallucination"`, my concern was that the *context* can alter the behavior of the original model, which can be transferred to the updated one *without context*

---

> ### Author Response · Authors · 2024-11-19
> **Response to Remaining Concerns (Part 1/2)**
>
> We would like to express our most sincere gratitude to Reviewer CZiX for their thorough and constructive feedback on our submission. Your detailed comments have been invaluable in helping us improve and clarify our work. Below, we address each of your remaining concerns:
>
> **[W1] Requirement of teacher LM**:
>
> We appreciate the reviewer's suggestions to make our statements clearer. We commit to do the following:
> - Add the discussion of FT (S) compared with FT (C) in our experiments, and acknowledge that there is some performance gain introduced by the Question-Answering pairs, potentially because the information is well-extracted and easy for the model to internalize during the fine-tuning process.
> - We are running the experiments of using `Mistral-7B-instruct-v0.3` to generate question-answering pairs for `Mistral-7B` in Table 1 and Table 2, with number of contexts being 100 and 500. After this, we will run experiments using `Llama3-8B-Instruct` to generate question-answering pairs for `Llama3-8B` in the similar settings. We will include all these results in our revised manuscript. This will serve as an important ablation study.
>
> We confirm that identical experimental settings were used for our proposed method and FT (S), Q. Specifically, they were trained using the same data, training settings, model configurations, and code implementation (the same code with only the loss value being calculated differently).
>
> For Single Context Injection and Batch Context Injection, the same generated QA pairs and Slimpajama data were used for both methods at every training step. The generated QA pairs and Slimpajama data were used in a 1:1 ratio. We trained the methods using the AdamW optimizer with a learning rate of $2\times10^{-5}$ for 50 epochs. The KL divergence was computed using the \texttt{torch.nn.functional.kl\_div} function from the PyTorch library. For the backbone model, OpenLLaMA-3B-v2, we trained the MLP layers. For Mistral-7B, LLaMA3-8B in Table 1 and Table 2, Mistral-7B-Instruct-v0.2 in Table 3, we use LoRA training with the `peft` package. The LoRA configurations were also the same for all these settings:
> ```
> {
>   inference_mode: false,
>   r: 8,
>   lora_alpha: 32,
>   lora_dropout: 0.1,
>   target_modules: ["q_proj", "v_proj", "k_proj", "up_proj", "down_proj", "gate_proj"] }
> ```
> We will add all these clarifications in our paper.
>
> **[W1-1] Cost implications of inference to augment context with QA pairs and KL fine-tuning:**
>
> - **QA-pair construction as preprocessing**: We acknowledge that describing QA-pair generation as a “preprocessing step” may have been misleading. While we intended to convey that this step is streamlined and straightforward, we recognize that it involves additional computational costs, which include two aspects: (1) Generating QA pairs using `gpt-4o-mini` (or an alternative instruct model) incurs additional inference costs; (2) Calculating the base logits with the original model $\theta$ for the calculation of KL divergence adds to the overall expense. Thus our cost (time cost and money cost) is about twice as the finetuning baseline. Specifically, there are four steps involved in our method: (a) generating the qa pairs, (b) calculating the base logits, (c) calculating our logits; (d) running backward and optimization. For the continual-finetuning baselines (i,e, `FT (C)` and `FT (S)` in Table 1-3), they involve (c) and (d). If we roughly regard (a)-(d) as similarly time-consuming, our cost is twice the cost of continual fine-tuning baseline. We will add these acknowledgments to our paper.
>
> - **Number of QA pairs:** We generate 50 QA pairs per context. This results in 100 training steps per context (50 for QA pair generation and 50 for SlimPajama data fine-tuning). During application, QA pair generation and fine-tuning can be batched, potentially reducing the number of steps required and improving efficiency. We did not use batch generation in our experiments as we only have A6000 and batch size larger than 2 is not affordable.
>
> - **Runtime Cost:** According to the reviewer's setting, our method would require 11 dollars to internalize the context. However, for finetuning baseline, it requires 5 dollars to finetune on the context. So basically we have doubled the cost. (We apologize for the potential misunderstanding, but compared with the finetuning baseline, it does not seem to have 10x cost to deploy).
>
> - **Summary of Costs**: We will revise the manuscript to fully acknowledge that our method introduces additional cost, making our cost about twice as the finetuning baseline, and we will add all these discussions in our paper. We would also like to argue that, given the effectiveness of our method, these costs might be acceptable, although we agree the efficiency remains a limitation and we will add clear clarifications in our Limitation section and try to mitigate these issues in our future work.

---

> > ### Author Response · Authors · 2024-11-19
> > **Response to Remaining Concerns (Part 2/2)**
> >
> > **[W3-2] Rapid integration:** We appreciate the reviewer’s suggestion to discuss the limitations of our method in the context of rapid integration. An important limitation is that **our approach does require time to train the model and internalize new contexts, which can introduce latency in user interactions**. This latency could negatively impact user experience, particularly in real-time conversational settings where prompt responses are essential. To address this limitation, some possible mitigation strategies can be introduced:
> >
> > - **Off-Peak Processing**: Context integration tasks can be scheduled during periods of low user activity, such as overnight or when user engagement is minimal. This ensures that model updates occur without disrupting ongoing user interactions.
> > - **Asynchronous Updates**: By decoupling the context integration process from the main user interaction flow, we can perform model updates asynchronously. This allows the system to continue responding to users while updates are being processed in the background.
> > - **Caching Mechanisms**: Implementing caching strategies for frequently accessed contexts can reduce the need for repeated QA pair generation and fine-tuning. By storing and reusing previously generated QA pairs for common contexts, we can accelerate the integration process for recurring information.
> >
> > We acknowledge that deploying these strategies in real-world applications involves additional considerations, such as varying conversation lengths and determining acceptable latency thresholds. Future work will explore these aspects in greater detail to enhance the practicality and responsiveness of our method in dynamic environments.
> >
> > **[W4] Hallucination**: We acknowledge the reviewer’s concern that our objective $Generate (\theta', p) \approx Generate(\theta | x, p)$ as shown in Eq.(1) will indeed affect the model's behavior as we are transferring the original model's ($\theta$) behavior after integrating $x$ into our new model $\theta'$. However, this seems to be an inevitable result if we want to internalize $x$ into model parameters and the effects from $x$ might be some desired property in our setting. We would love to hear more from the reviewer if we have misunderstood anything.
> >
> > **Summary of the Rebuttal**:
> >
> > We commit to addressing the reviewers' feedback by implementing the following revisions:
> >
> > 1. **Clarified Discussions**:
> >    We will enhance the manuscript with clearer discussions regarding the comparison between `FT (C)` and `FT (S)`, as well as the performance improvements gained from the constructed question-answering pairs.
> >
> > 2. **Additional Experiments**:
> >    We will include an ablation study evaluating the effectiveness of our method using the `Mistral-Instruct-V0.3` and `Llama3-8B-Instruct` models to provide a broader analysis.
> >
> > 3. **Cost Analysis**:
> >    We will acknowledge and provide a detailed discussion on the additional computational costs incurred by our approach, which approximately doubles the cost compared to continual fine-tuning.
> >
> > 4. **Limitations and Mitigation**:
> >    We will explicitly address potential challenges when deploying our method in practical settings, such as ChatGPT-like user services, and propose strategies to mitigate these limitations.

---

> ### Comment · Reviewer_CZiX · 2024-11-20
>
> Thank you for the clarifications.
>
> - [W1] ✅ Concerns with external teacher requirements are resolved. I'm convinced that the QA generation can be done by the model (I assume there is a minimum bar of model capability, but 7B seems to suffice).
> - [W1-1] ☑️ Runtime costs are significant but clearly acknowledged by authors. **Despite substantial runtime costs, the method demonstrates a marked improvement over previous approaches (discussed in the paper) in preserving context, presenting a promising direction to solve this problem.**
>   - Given that 50 QA pairs were generated and use to fine-tune the model, the runtime cost is significant–generating 50 QA pairs and fine-tuning the model using these, compared to the cost of processing the context by the base model. In my analogy, it would cost $500 dollars, or 500x to deploy in a ChatGPT-like service, as-is. It is also likely significantly more expensive than other non-fine-tuning baselines such as RAG. As authors mentioned, the training part is 2x more expensive than vanilla fine-tuning on the QA pairs, `FT (S)`.
> - [W3] ☑️ Limitations in rapid integration are clearly acknowledged by authors, and there are many avenues to solve them.
> - [W4] ✅ My concerns with hallucinations are resolved. I agree with authors on the fundamental limitations, and these have been adequately addressed.
>
> I raise my overall score from 3 to 6 and contribution from 2 to 3. Please revise the manuscript to accommodate the numerous changes discussed during the rebuttal for the final version.

---

> > ### Author Response · Authors · 2024-11-21
> > **Response to Reviewer CZiX**
> >
> > We sincerely thank Reviewer CZiX for their thoughtful follow-up feedback, which has been invaluable in improving and clarifying our work.
> >
> > Regarding [W1-1], we acknowledge that our method incurs additional costs during knowledge injection compared to using context directly with the model or employing RAG. However, it offers distinct advantages by leaving the model's test-phase operational requirements unchanged. Specifically, the updated model requires no additional storage, run-time memory, computational overhead, or latency, and retains the same context window size as the original. In scalable applications, test-phase requirements often account for a significant portion of total costs due to a high volume of users and frequent queries. These factors, such as latency, can also critically impact the overall user experience.
> >
> > In contrast, approaches such as **direct context usage** or **RAG** demand additional storage for maintaining contexts, increased run-time memory and computational resources to process them, and added latency for retrieval. They also reduce the effective context window size available for text generation.
> >
> > Compared to **fine-tuning baselines**, our method doubles the training-phase cost but maintains identical test-phase costs. Moreover, our method demonstrates significantly higher performance in incorporating new knowledge and retaining past knowledge.
> >
> > In summary, our method has unique properties compared to direct context usage and RAG methods, while also achieving significant improvements over fine-tuning baselines.
> >
> > Once again, we sincerely thank the reviewer for providing the insightful feedback, which has provided invaluable guidance in improving our work.

---

### Official Review · Reviewer_wRpc · 2024-11-03

**Soundness:** 2
**Presentation:** 3
**Contribution:** 2
**Rating:** 6
**Confidence:** 4

**Summary:**

This paper introduces SELF-PARAM, a novel method for updating LLMs with new knowledge without requiring additional storage parameters. The key insight is using KL divergence minimization between models with and without context access to embed knowledge directly into model parameters. Through comprehensive experiments across various injection scenarios and a practical demonstration on conversational recommendation tasks, the authors show SELF-PARAM outperforms existing methods while maintaining zero storage overhead - effectively solving the challenge of efficient knowledge integration in LLMs.

**Strengths:**

1. The writing of the paper is clear and easy to follow and understand.
2. The paper proposes a novel and elegant solution to knowledge injection in LLMs by using KL divergence minimization - it's technically sound while being simple. The zero additional storage requirement is a plus compared to existing methods that need external memory or retrieval modules.
3. The empirical evaluation is overall comprehensive and convincing. The authors tested their method across multiple dimensions (single/batch/sequential injection) and backed up their claims with solid results.

**Weaknesses:**

1. The computing cost is significantly larger for the proposed method compared to next-token-prediction fine-tuning. For one query, it has to run the base model twice to get the two corresponding probability distributions. This paper only discusses the space complexity. The comparison of time complexity is also worth discussing.
2. In the experiment setting, this method utilizes two popular dense retrieval (DPR) and spare retrieval (BM25) methods as baselines. However, these two methods are a little bit too old to be considered as the baselines. Using a stronger retrieval model, such as [1][2], will better demonstrate the superiority of the proposed method.

[1] Izacard, Gautier, et al. "Unsupervised dense information retrieval with contrastive learning." arXiv preprint arXiv:2112.09118 (2021).

[2] Wang, Liang, et al. "Text embeddings by weakly-supervised contrastive pre-training." arXiv preprint arXiv:2212.03533 (2022).

**Questions:**

1. I found the term self-updating very confusing here. This up-updating method requires (1) an extra training process to update the parameters and (2) an extra instruct model (GPT-4o-mini) to construct the training data. I think the authors could provide more insight into how to understand self-updating.
2. Retention is discussed in the introduction but not mentioned in the experiments. I wonder how the authors evaluate the aspects of retention performance.
3. An ablation study is missing from this paper; there are multiple techniques (KL-div, diverse instructions construction...) introduced in this method, and how each one contributes to the final result should be investigated and compared. At least, some discussion should be included.

---

> ### Author Response · Authors · 2024-11-15
> **Rebuttal to Reviewer wRpc**
>
> We sincerely thank the reviewer for the insightful suggestions! We address the concerns as below:
>
> **[W1] Computation Cost**: We appreciate the reviewer’s concern regarding computation cost and acknowledge that our method involves an additional forward pass. However, we would like to clarify that the backward pass is comparable to standard fine-tuning with a next-word-prediction loss. This yields a time complexity ratio of 2:3 between our method and standard fine-tuning, assuming equal forward and backward pass times (though the backward pass can be slower). While our method indeed incurs additional computation, we believe this overhead is justified by the performance gains it achieves, which standard fine-tuning would not match regardless of training duration.
>
> **[W2] Baselines**: We acknowledge that BM25 and DPR are less recent retrieval methods, and appreciate the reviewer’s suggestions. We would like to mention that we have also included RAPTOR as a baseline, which is a current approach (published in January 2024) and actually newer than the recommended methods (published in 2021 and 2022). Nonetheless, we will incorporate these suggested references into the related work section.
>
> **[Q1] Clarification of Self-Updating**: By “self-updating,” we refer to the model’s capability to integrate new data autonomously. In our experiments, we utilized `gpt-4o-mini` to generate the Target Sentence Set, but this could be replaced by the model itself if the model is fairly powerful, such as `mistral-7b-instruct`. Once we obtain the Target Sentence Set, the training is conducted solely on the model itself. In practical applications, APIs such as `gpt-4o-mini` may be used to preprocess incoming data before self-updating. Here, the role of APIs serves primarily as preprocessing rather than altering the model’s inherent updating capability. We will further clarify this terminology to prevent ambiguity.
>
> **[Q2] Evaluation of Retention**: Thank you for highlighting the need for clarity regarding retention evaluation. In Section 4.3, we describe how sequential injection experiments are used to assess retention, specifically under “Results and Discussions.” To address this, we will add an explicit statement at the beginning of the Experiment section to ensure the retention evaluation setup is clear.
>
> **[Q3] Ablation Study**: We appreciate the reviewer’s suggestion to further clarify our ablation analysis. Our study incorporates two main techniques: (1) KL-Divergence minimization (Eq. 2) and (2) Target Sentence Set construction. In our experiments, we conducted the following analyses:
> 	- We fine-tuned the model solely on the Target Sentence Set, labeled as `FT (S), Q` in Table 1-2 and `FT (Q)` in Table 3. This setting serves as an ablation for KL-Divergence, isolating the effect of our objective function by comparing the results of `FT (S), Q` with those of our full method. This comparison highlights the unique contribution of KL-Divergence to our approach.
> 	- Further, comparing `FT (S), Q` with `FT (C), Q` (fine-tuning on context without the Target Sentence Set) allows us to assess the impact of the Target Sentence Set. The performance gains of `FT (S), Q` over `FT (C), Q` indicate that the Target Sentence Set offers marginal improvements, but the primary performance boost is attributed to our design in Eq. (2).

---

> ### Comment · Reviewer_wRpc · 2024-11-25
>
> Thank you for your rebuttal. I have no further questions.

---

### Meta-Review · Area_Chair_t5w2 · 2024-12-21

**Metareview:**

The authors introduce SELF-PARAM, a novel parameter- and storage-free approach designed to enhance language models with the ability to retain long-term knowledge while effectively incorporating recent events. The method leverages an instruction-following model to generate contextually relevant question-answer pairs, subsequently training the target model by minimizing the KL divergence between its output distribution with and without the provided context.

A key point raised by reviewers CZiX and 3X9B is the similarity of SELF-PARAM to existing LLM distillation techniques, given its reliance on an external LLM for context augmentation. While the paper focuses on memory injection and retention, a more comprehensive discussion of related work would strengthen the contribution. Specifically, exploring research on leveraging external LLMs to enhance target LLM capabilities is warranted. Examples include work on distilling knowledge from larger models [A1] and aligning models with synthetic data [A2]. Incorporating this discussion would provide valuable context and highlight the distinct aspects of the proposed approach.

[A1] Distilling Step-by-Step! Outperforming Larger Language Models with Less Training Data and Smaller Model Sizes. ACL 2023
[A2] CodecLM: Aligning Language Models with Tailored Synthetic Data. NAACL 2024

**Additional Comments On Reviewer Discussion:**

Reviewers initially raised concerns regarding computational cost, baseline comparisons, the clarity of the "self-updating" mechanism, and sensitivity to the diversity of the generated question-answer pairs. Following the authors' rebuttal, reviewers expressed satisfaction that these concerns had been adequately addressed.

---

### Decision · Program_Chairs · 2025-01-22

Accept (Poster)